# First Nationwide Analysis of Riming Using Vertical Observations from the Operational German C-Band Radar Network

Paul Ockenfuß<sup>1</sup>, Michael Frech<sup>2</sup>, Mathias Gergely<sup>2</sup>, and Stefan Kneifel<sup>1</sup>

Correspondence: Paul Ockenfuß (paul.ockenfuss@physik.uni-muenchen.de)

Abstract. The 17 operational German C-band polarimetric weather radars routinely perform a vertical "birdbath" scan, which has so far primarily been used for calibration of differential moments. In this study, we transfer a retrieval algorithm for the rime fraction of snowflakes - originally developed for Ka-band cloud research radars - to the operational birdbath scan. This retrieval, which relies on the increase in detected mean Doppler velocity, serves as our benchmark. To validate the transfer of the retrieval, we apply it to a "mockup" birdbath dataset, constructed by downsampling cloud radar data to match the resolution of the operational birdbath scan. In addition, we present a new clutter filter and a melting layer detection algorithm for the operational birdbath scan. Finding good agreement between mockup and benchmark datasets, we apply the new retrieval to radar data recorded during the winters of 2021 to 2024. This results in a nationwide map of riming events in wintertime clouds. There is a north-south gradient in the riming distribution, which can be linked to Germany's precipitation climatology. Notably, we show that the occurrence of riming events correlates more strongly with precipitation intensity than with the total number of precipitation hours across sites. The temperature distribution associated with riming is consistently between  $-15\,^{\circ}\mathrm{C}$  and  $0\,^{\circ}\mathrm{C}$  at all sites, except for the Feldberg site, which hints at a possible orographic effect. This study demonstrates that the operational birdbath scan of C-Band weather radars can be used for the retrieval of microphysical processes. Corresponding solutions, challenges and methods to transfer retrieval algorithms from research cloud radars to the operational weather radars are discussed.

## 1 Introduction

In Germany, the national weather radar network operated by the German Weather Service (DWD) comprises 17 polarimetric C-band radars. While these operational radars primarily perform wide area azimuth scans (commonly referred to as plan position indicator (PPI) scans) at 10 elevations, they also include a short vertical scan at zenith as part of their scanning routine. Such vertically pointing scans, known as "birdbath scans," have long been a standard component in the measurement strategy of cloud research radars, where they support detailed investigations of cloud and precipitation microphysics (Illingworth et al., 2007).

In contrast, operational weather radars have historically used the birdbath scan mainly for calibration purposes (Frech et al., 2017). Its scientific potential remains largely untapped, with only a few recent studies beginning to explore its capabilities. For

<sup>&</sup>lt;sup>1</sup>Meteorologisches Institut, Ludwig-Maximilians-Universität München, Germany

<sup>&</sup>lt;sup>2</sup>German Meteorological Service (Deutscher Wetterdienst, DWD), Observatorium Hohenpeißenberg, Hohenpeißenberg, Germany

35

instance, Frech and Steinert (2015) analyzed a strong rain event using birdbath scan data, Gergely et al. (2025) derived hail size distributions from Doppler spectra of three hail cases, and Blanke et al. (2025) identified six cases of strong riming based on mean Doppler velocities at the Essen radar site.

In this publication, we present the first long-term scientific analysis of birdbath scan data collected from all 17 C-band radar sites. We focus on the detection and characterization of rimed particles. Riming refers to the collision of frozen hydrometeors—such as ice crystals or aggregates—with supercooled liquid water droplets in mixed-phase clouds. Upon impact, the droplets freeze, causing the hydrometeors to gain mass (Lasher-Trapp, 2022; Pruppacher and Klett, 1996). This process removes liquid water from the cloud and contributes efficiently to precipitation formation (Grazioli et al., 2015; Houze and Medina, 2005; DeLaFrance et al., 2024). However, modeling riming accurately remains challenging due to the complex phase interactions and irregular shapes of frozen hydrometeors (Leinonen and Szyrmer, 2015; DeLaFrance et al., 2024).

One of the most robust indicators of riming is an increased sedimentation velocity of the hydrometeors, resulting from their higher bulk density compared to unrimed ice crystals or aggregates. In vertically pointing radar observations, this manifests as enhanced mean Doppler velocities, which can be used to classify the presence and intensity of riming (Mosimann, 1995; Kneifel and Moisseev, 2020). Strong riming typically occurs in the form of discrete events, which happen on a multi year average every fourth to fifth day and can last from several minutes to several hours (Ockenfuß et al., 2025). Therefore, short-term measurement campaigns often fail to capture a sufficient number of riming events for robust statistical analysis.

The operational C-band radar network, with its long-term, continuous measurements and equidistant spatial coverage, provides an ideal platform for investigating phenomena such as strong riming on a larger spatial and temporal scale. For the first time, it enables the study of the spatial variability of riming across a contiguous region.

This publication has two main objectives. Firstly, we develop a riming detection algorithm tailored to the C-band birdbath scan, building upon the approach presented in (Ockenfuß et al., 2025) for Ka-band cloud radars. This involves two new processing components: a clutter filter and a melting layer detection algorithm specifically designed for the C-band birdbath scan. Both are computationally efficient and operate without the need for manual intervention, making them suitable for large-scale, multi-year datasets. The resulting detection algorithm enables us to quantify and compare the frequency and temperature dependence of riming across all 17 radar sites. We further relate riming frequency to local surface precipitation climatologies. Secondly, we use the riming detector for highlighting the broader challenges involved in transferring retrieval algorithms from research cloud radars to operational radar systems. We present solutions to these challenges that should be applicable beyond the specific application of riming detection.

The remainder of this publication is organized as follows: section 2 describes the datasets and the methodology used to adapt the riming retrieval to the C-band radars. In section 3, we first evaluate the performance of the new retrieval and then present the spatial and temporal distribution of riming across Germany. Characteristics and uncertainties of this distribution are discussed in section 4. section 5 concludes the study.

### 2 Methods

## 2.1 Benchmark Cloud Radar Riming Retrieval

Ockenfuß et al. (2025) developed a riming detection algorithm for observations from Ka-band cloud radars like the MIRA35 35 GHz cloud radar located at the Lindenberg observatory, Germany, and operated by the German Weather Service (Görsdorf et al., 2015). The data is accessed via the Cloudnet database (Illingworth et al., 2007), with a time resolution of 30 s and a height resolution of 35 m. The retrieval involves several filtering steps in order to separate the rain and ice part of the cloud and exclude convective regions with significant vertical air motion. Table 1 gives an overview of those steps. Afterwards, a unique relation between rime mass fraction (FR) and radar Doppler velocity from Kneifel and Moisseev (2020) is applied to the ice part of the cloud. FR is defined as

$$FR = \frac{m_{\rm r}}{m_{\rm tot}} = 1 - \frac{m_{\rm c}}{m_{\rm tot}}$$

with  $m_{\text{tot}}$  the total mass of the particle,  $m_r$  the rime mass and  $m_c$  the mass of the unrimed crystal. The minimum detectable FR with this method is 0.6, corresponding to  $1.5 \,\mathrm{m\,s^{-1}}$  Doppler velocity in Ka-band radars. In a last step, nearby profiles with riming detections are clustered into connected riming events, based on a density clustering scheme described in Ockenfuß et al. (2025). Figure 7 a and b show two examples of stratiform, frontal systems with embedded riming, as well as the detected riming regions and the event clustering. The first example is a widespread precipitation system passing over Germany from the Southeast. This synoptic situation is often associated with intense precipitation. In this case, we see very strong riming signatures over an extended period of time. The second example also depicts a frontal passing, but with smaller riming cells, more typical for the majority of events. In the following, we will reference the cloud radar algorithm and the corresponding results as the "benchmark retrieval" and "benchmark results", respectively.

#### 2.2 The operational C-Band Birdbath Scan

The German Weather Service (DWD) operates 17 C-band radars spread equidistantly across Germany. Their locations are depicted in Figure 1. At each site, a birdbath scan is performed every 5 min as part of the operational scanning cycle. Originally, it was introduced to calibrate differential reflectivity ZDR (?). While the radar is oriented vertically, the dish is turning in the azimuth direction in order to smooth possible orientational effects in the radar data. The full scan takes 15 s, consisting of 15 "rays" of 1 s duration. Per ray, the radar moments are computed from a batch of 1024 pulses and stored on disk. Since July 2021, for each ray the full Doppler spectra are stored in a separate file. In the vertical, the intrinsic data resolution is 60 m, but sampling is performed at 25 m resolution (Gergely et al., 2022). In the following, we will reference the retrieval and results based upon this dataset as the "operational retrieval" and "operational results", respectively

## 2.3 Retrieval Transfer

Based on our experience, when transferring any retrieval originally developed for cloud research radars to the operational C-band birdbath scan, three universal key points need to be adressed:

**Figure 1.** Overview map of Germany with locations of all measurement devices used in this publication: the 17 operational C-Band radars, the cloud radar in Lindenberg, the sounding stations and the surface weather stations.






- 1. Radar band: While cloud radars typically operate in Ka-band (35 GHz) or W-band (94 GHz), the operational radars in Germany work at C-band (5 GHz). Therefore, compared to cloud radars, they are less affected by attenuation due to hydrometeors and gases. At the same time, as we show in subsection 2.4, clutter limits the usable reflectivity range of the C-band birdbath scans to values higher than  $-20 \, \mathrm{dB}$ , which is less than the  $-40 \, \mathrm{dB}$  typically obtained by vertically looking cloud radars. As a consequence, for example low reflectivity cloud tops are not visible in the C-band birdbath scan (Frech et al., 2025). The same is true for liquid water peaks around  $0 \, \mathrm{m \, s^{-1}}$  fall velocity in the Doppler spectra (Gergely et al., 2022).
  - 2. Time resolution: The vertical resolution of 25 m of the C-band radar is comparable to the typical resolution of 30 m to 40 m of Ka-band radars in the Cloudnet database, but the time resolution of one scan every 5 min is much coarser than typical cloud radar sampling rates. In Cloudnet, cloud radar data has a time resolution of 30 s. Therefore, all operations which act along the time dimension must be reevaluated and retuned when applied to C-band birdbath data. This affects common operations like time domain filters, fallstreak tracking (e.g. Kalesse et al. (2016)) or eddy dissipation rate retrievals (e.g. Borque et al. (2016)).
  - 3. Additional data: Most modern retrieval techniques depend on more than just radar data. Surface weather stations, equiped with standard meteorological instrumentation, can provide information about the local weather conditions and the local climatology. Additional active and passive remote sensing instruments like lidars or microwave radiometers can complement radar observations, e.g. to detect liquid layers in ice clouds. High resolution model profiles are helpful for cross-checking and interpreting measurement results. For most research sites in Cloudnet, those data sources are available, but currently there is almost no additional instrumentation at the DWD C-band radar stations. Even vertically resolved model profiles are scarce. Model reanalysis datasets like ERA5 (Hersbach et al., 2020) are usually stored in a way which favors spatial analysis over the download of long, single-location time series, which would be required to complement birdbath observations.

Different strategies can be applied to overcome these issues. For the last point, some of the external information can be derived from the radar, e.g. the melting layer height (subsection 2.5). In other cases, climatologies can replace local measurements, e.g. in case of the pressure profile (subsection 2.7). To quantify the influence of point two, the difference in resolution, we create a "mockup" C-band radar dataset from cloud radar data. As listed in Table 1, this mockup dataset is created from the cloud radar data by taking only every 10th profile and upsampling the vertical to the same height levels as the operational C-band birdbath scan. Comparing the results of filters and algorithms on the mockup dataset with the results from the benchmark retrieval, we can asses if they are directly transferable or need readjustment. We will use this strategy for the convection filtering (subsection 2.6) and event detection (sec results/statistical comparison) part of the retrieval. Point one, the difference in radar frequency, is difficult to generalize. In our case, it is addressed by simulations for different wavelengths from Kneifel and Moisseev (2020). They simulated FR-MDV relationships for the X-, Ka- and W-band (corresponding to  $10 \, \mathrm{GHz}$ ,  $35 \, \mathrm{GHz}$  and  $94 \, \mathrm{GHz}$ , respectively). Differences between those simulations arise if some hydrometeors within the measurement volume

|                          | Benchmark Retrieval (Ock-     | Mockup Retrieval                | Operational Retrieval                     |
|--------------------------|-------------------------------|---------------------------------|-------------------------------------------|
|                          | enfuß et al. (2025))          |                                 |                                           |
| Data source              | Ka-band cloud radar           | Ka-band cloud radar             | C-band Operational Radar                  |
| Time resolution          | 30 s                          | 5 min                           | 5 min                                     |
| Vertical Resolution      | 35 m                          | 25 m                            | 25 m                                      |
| Blind range              | 250 m                         | 250 m                           | 600 m                                     |
| Melting Layer Detection  | Based on Cloudnet Classifica- | Retrieved from Radar Doppler    | Retrieved from Radar Doppler velocity     |
|                          | tion                          | velocity                        |                                           |
| Fall Velocity Correction | Hourly, site specific model   | Climatological pressure profile | Climatological pressure profile           |
|                          | pressure profiles             |                                 |                                           |
| Convection Filtering     | Mosimann1995 index based on   | Mosimann1995 index based on     | - Mosimann1995 index based on 5 data      |
|                          | 40 datapoints                 | 5 data points                   | points                                    |
|                          |                               |                                 | - High rain rate (reflectivity) threshold |
| Temperature Filtering    | Wetbulb Temperature from      | No filtering                    | No filtering                              |
|                          | model profiles                |                                 |                                           |

Table 1. Comparison of retrieval configurations for the "benchmark", "mockup" and "operational" retrieval.

are large enough to exhibit Rayleigh scattering in one band and Mie scattering in another band. However, for the combination of X- and C-band, this transition size is above 1 cm and Rayleigh scattering can be assumed for both bands (Matrosov, 1992).

## 115 2.4 Clutter Filter




When looking slant, the sensitivity of the operational radars is  $-36 \,\mathrm{dB}$  at  $1 \,\mathrm{km}$  range. For the birdbath observations, such sensitivities can not be achieved, since clutter echos dominate over weakly reflecting meteorological targets. These clutter signals are created when antenna sidelobes hit surrounding ground targets. In the radar signal processor, each range gate is quality controlled using thresholds for the noise  $(-9 \, dB)$ , the clutter power  $(-25 \, dB)$  and the cross polar correlation coefficient (0.45). This thresholding is applied to all unfiltered moments in every range gate of the 15 rays. Figure 2a shows only the 'valid' observations, i.e. the observations where all 15 contributing samples passed the quality control. As is evident from the horizontal clutter lines in the scene, the signal processor filtering is not sufficient. The number and height of the clutter lines varies for each radar site, depending on the ground targets in the surrounding of the radar. The reflectivity of the lines is usually between  $-20 \, dB$  to  $-10 \, dB$ . In the 2D reflectivity histogram in Figure 2c, the clutter is visible in the form of distinct patterns at the lower reflectivity edge. In order to remove the clutter, while keeping as much of the weather signal as possible, a height and site dependent reflectivity threshold filter was developed. It makes use of the fact that every birdbath observation consists of 15 independently stored samples. For every height level, we determine the typical reflectivity of the "invalid" observations, i.e. the observations where the signal processor flagged at least one ray as invalid, based on the criteria described above. Specifically, we take the upper 5% reflectivity quantile of all invalid observations. We then apply this reflectivity as a threshold to the valid observations. From Figure 2c, it is evident that the height dependent reflectivity threshold is able to separate the clutter and weather parts in the histogram well for most height range. It can be seen that with this filtering, depending on the clutter

Figure 2. (a) Example of a raw measurement at the Essen radar site, with only basic filtering by signal processor. (b) The same scene after clutter filtering, where the clutter is removed based on a height dependent reflectivity threshold. (c) 2D reflectivity by altitude histogram. The clutter lines are visible as horizontal stripes in the histogram. The black line indicates the height dependent reflectivity threshold.

structure, the operational birdbath scan can detect weather signals down to  $-20 \, \mathrm{dB}$  in a range from around  $1 \, \mathrm{km}$  to  $10 \, \mathrm{km}$  above the radar. The example scene after discarding the clutter part from the histogram can be seen in Figure 2b. Since the lowest range bins are almost always dominated by strong clutter, we define a minimum valid range of  $600 \, \mathrm{m}$  at all radar sites. Compared to other methods for clutter filtering, e.g. the method presented by Gergely et al. (2022) based on Doppler spectra, our method requires no human judgement and has very low computational cost. Therefore, it is well suited to filter multiple years of radar moments from all 17 German radar sites.

## 2.5 Melting Layer Detection



In order to detect rimed particles in the ice phase of the cloud, a reliable melting layer detection is necessary. In the cloud radar retrieval, this was achieved based on the Cloudnet target categorization (?) and vertical temperature profiles from model reanalysis. Since neither target categorization nor model profiles are available at the C-band radar sites, we need to derive the melting layer from the C-band radar. For this task, the retrieval needs to work robustly on multiple sites, ideally without site-specific retuning of parameters. We do not require meter accuray, but we want to avoid severe misdetections, where the melting layer is placed too low into the warm part of the cloud. Due to the high velocities of rain drops, those cases would falsely be detected as riming. Since actual riming is comparably rare, this would bias the frequency statistics of riming events.

Usually, melting layer detection is done by detecting peaks in the vertical profiles of radar moments, e.g. radar reflectivity

Usually, melting layer detection is done by detecting peaks in the vertical profiles of radar moments, e.g. radar reflectivity and correlation coefficient as in Sanchez-Rivas and Rico-Ramirez (2021). In our case, we use Doppler velocity as the basis for the retrieval. Doppler velocity is unaffected by attenuation, can be measured with high accuracy and has a direct physical interpretation with high velocities in the rain part (usually  $>2 \text{ m s}^{-1}$ ) and lower in the ice part ( $



of strong riming. Additionally, it allows us to apply the retrieval with the same parameters to real C-band data and the mockup dataset based on Ka-band data.

From the example in Figure 3a, we see that in Doppler images, the melting layer is characterized by two properties:

- A sudden increase in fall velocity towards the ground: As in Sanchez-Rivas and Rico-Ramirez (2021), we search for maxima in the vertical gradient  $\partial_h V(h)$  of Doppler velocity V(h), but we use the 3x3 Sobel filter presented in Wolfensberger et al. (2015) to calculate the gradients. From manual analysis, we found  $0.2 \,\mathrm{m\,s^{-1}\,m^{-1}}$  to be a good gradient threshold for potential locations of the melting layer.
- Faster Fallspeeds below the melting layer than above: Usually, the melting layer separates the Doppler image clearly into a slow and fast falling part (compare Figure 3a). Therefore, we propose the difference in average velocities  $\Delta_h V(h)$  below and above the potential melting layer as a novel, additional criteria:

160 
$$\Delta_h V(h) = \frac{1}{h} \int_0^h V(x) \, dx - \frac{1}{H - h} \int_h^H V(x) \, dx \tag{1}$$

Here, h is the position along the vertical and the cloud extends from height 0 to H. This concept is illustrated in Figure 3b.

The melting layer is determined as the level with the maximum gradient  $\partial_h V(h)$ , weighted by  $\Delta_h V(h)$ :

$$h_{\text{meltinglayer}} = \underset{h}{\arg\max} (\partial_h V(h) \cdot \Delta_h V(h)) \tag{2}$$

Including  $\Delta_h V(h)$  brings information about the full column into the retrieval, instead of looking only at localized gradients. This makes it much less likely that a sudden, local fluctuation in Doppler velocity, e.g. due to riming or cloud top turbulence, is misinterpreted as the melting layer. For our application, no retrieved melting level is better than a wrong melting level. Therefore, in a postprocessing step, we remove all points where the melting layer appears to be changing by more than  $300\,\mathrm{m}$  per  $5\,\mathrm{min}$ . In Figure 3c, we compare with the radar derived melting layer with the melting layer from radiosonde measurements at the Essen site, where a sounding station directly next to the radar performs two soundings per day. We define the radiosonde melting layer as the highest level, where the radiosonde crosses a zero degree isotherm of wetbulb temperature. Wetbulb temperature is calculated from radiosonde pressure, temperature and relative humidity using Normand's method (Knox et al., 2017). We see that the retrieved melting level follows the radiosonde melting layer very consistently with an offset of  $200\,\mathrm{m}$  ( $\pm 50\,\mathrm{m}$ ). Such an offset is to be expected, since, depending on the humidity, snowflakes can sustain positive temperatures up to  $+4\,^{\circ}\mathrm{C}$  on average (Heymsfield et al., 2021). In Fig. 2c, only winter months (Nov. to Apr.) are take into account, and only cases where a melting level was detected in the radar as well as in the radiosonde within less than  $1\,\mathrm{h}$ . For the riming detection, we rely on the radar derived melting layer (plus  $200\,\mathrm{m}$ ) whenever possible. Only if no reliable detection was possible, we extrapolate the last value up to  $1\,\mathrm{h}$  and otherwise use the value from the nearest sounding station as a proxy. In the latter case, at most  $12\,\mathrm{h}$  are tolerated between sounding and radar profile.

**Figure 3.** (a) Example of a large scale, precipitating system with signatures of riming embedded. The black line denotes the detected melting layer, the red line shows the time of the profile in panel b). (b) Example of a vertical profile of mean Doppler velocity. The background colors separate the image in two parts, maximizing the difference in average Doppler velocity between the parts. (c) Comparison of the radar derived melting layer with the radiosonde derived melting layer at the Essen radar site

180

185

190

195

200

## 2.6 Convection Filtering

For microphysical retrievals based on MDV, one has to make sure that the signal variability is dominated by cloud microphysics and not vertical air motions. In order to detect convective areas and regions with wave activity in clouds, Mosimann (1995) proposed the so-called convection index  $\kappa$ . The convection index is essentially the statistical coefficient of variation (standard deviation divided by mean) of MDV.

$$\kappa = \frac{|\text{MDV}(z) - \overline{\text{MDV}(z)}|}{\overline{\text{MDV}(z)}}$$

 $\overline{\text{MDV}(z)}$  is the average MDV in a  $\pm 10\,\mathrm{min}$  rolling window. It is assumed that positive MDV indicates downward motion. Mosimann (1995) established  $0 

**Figure 4.** (a) The Mosimann convection filter applied to a summer precipitation event in the original 30 s resolution. (b) Convection filter applied to the event from (a) in 5 min resolution. (c) Comparison of the filtered areas in (a) and (b).

by Heymsfield et al. (2013):

$$\begin{aligned} \text{MDV}_{\text{surface}} &= \text{MDV} * \eta \\ \eta &= \left(\frac{p}{p_{\text{surface}}}\right)^{0.4} \end{aligned}$$

For the pressure profile, Ockenfuß et al. (2025) used hourly profiles from numerical weather models. In Blanke et al. (2025), colocated radiosonde soundings were used for this task. For the 17 C-band sites, neither model nor radiosondes are directly available at all sites. However, we argue that the site and time specific pressure profiles can be replaced by a generic climatological profile. Figure 5a shows the variability  $\sigma_{\eta}$  of  $\eta$  for different heights, relative to the climatological mean.

$$\sigma_{\eta} = \frac{\eta - <\eta>}{<\eta>}$$

 $<\eta>$  denotes the climatological average over 20.000 soundings from the Lindenberg observatory between the years 2010 to 2024. As is evident, the error in  $\eta$  by neglecting the temporal variability in atmospheric pressure is below 1% for almost

**Figure 5.** (a) Relative variability of the pressure correction factor due to natural pressure variability. Colors show the frequency in arbitrary units. (b) Example of a convective event at the Essen radar site. The red time interval is excluded from any analysis due to the high reflectivities in the liquid part and the high Doppler velocities in the ice part of the cloud. The black line shows the extrapolated melting layer, detected by the method described in subsection 2.5.

all cases within the typical height ranges of riming  $(0 \,\mathrm{km}$  to  $6 \,\mathrm{km})$ . This error is smaller than the uncertainty inherent in the MDV-FR relation due to the variability of the underlying measurements by Kneifel and Moisseev (2020), and can therefore be neglected.

## 2.8 Clustering

The time intervals where rimed particles are detected are then clustered into riming events. For this step, we use the same definitions and parameters as in Ockenfuß et al. (2025). They define a riming event as the maximum time interval, in which at least 75% percent of the birdbath profiles show riming in at least one range gate. Riming events covering less than  $2 \min \cdot \ker$  in the time-height image are discarded as noise. This translates to events consisting of less than 17 time-height samples for the C-band birdbath scan.

## 215 2.9 Histogram Correction

In order to compare the temperature distribution of riming events between different radar sites in subsection 3.4, we need to correct the distributions for deviations due to the following three points:

1. Differences in the radar uptime: Due to technical problems or maintenance, the total number of available observations varies. Radars with less uptime naturally detect less riming cases.

Figure 6. Number of observations per isotherm and radar.

- 2. Differences in radar elevation: Radars located at higher elevations detect less riming at warmer temperatures than radars at lower elevations, since higher elevation radars can only observe the upper part of the atmosphere.
  - 3. Differences in riming climatology: There are natural differences in the total number of riming detections between the sites. Those differences will be analyzed in detail in subsection 3.3. We will correct for them for the temperature analysis in subsection 3.4.
- In order to correct for the points above, we count how often a certain isotherm is observable for each radar, taking into account the radar elevation and blind range. The isotherm information is again taken from the closest radiosonde to each radar site. We require less than 12 h difference between radar and radiosonde observations. Figure 6 shows an example of the number of observations per isotherm for the Feldberg, Memmingen and Isen sites during winter. We clearly see that the Memmingen radar had a longer maintenance period and therefore less observations in total. We also see that the Feldberg radar is located at a higher elevation and therefore, there are less observations at warmer temperatures compared to the Isen radar. For the results in subsection 3.4, we divide the absolute number of riming detections per radar and temperature level by the curves in Figure 6. This corrects for the points 1 and 2. Then, normalizing the resulting distributions to have an area of one under the curve corrects for point 3.

**Figure 7.** Examples of two events with riming signatures. Left column: April 14, 2023. Right column: April 28, 2023. First row ((a) and (b)): Observations by the Lindenberg cloud radar in 30 s resolution, with riming regions detected by the benchmark retrieval. Second row ((c) and (d)): Mockup C-band radar data in 5 min resolution, based on the data from row one. Riming regions detected by the mockup retrieval. Third row ((e) and (f)): C-band radar data in 5 min resolution from the Prötzel radar site. Riming regions detected by the operational retrieval.

## 3 Results


## 3.1 Comparison with Mockup Data

Figure 7 c and d show the C-band mockup version of the cloud radar measurements in Figure 7 a and b. As we can see, the large scale features of the cloud system are preserved well even with only 5 min resolution. In general, the newly developed retrieval applied to the mockup dataset is able to highlight the same regions within the cloud as the benchmark retrieval. For





Figure 8. (a) Monthly number of riming events detected by the benchmark and mockup retrieval. (b) Same as (a), but for the total monthly duration of riming.

some cases, the change in resolution leads to a different, but still reasonable, choice of the event clustering algorithm. Figure 8 a and b compares the benchmark and mockup retrieval statistically with respect to the monthly number of events (Figure 8a) and the monthly duration (Figure 8b). We see that the mockup retrieval can produce slightly different results for some months, but the overall correlation between both retrievals is good. On average, the mockup retrieval seems to detect a little bit more/longer riming events per month, as is evident from the red fit line being above the black 1:1 line. From manual inspection, we found multiple reasons for this: Sometimes, as for example in Figure 7 a and c or Figure 7 b and d, the clustering can be different. In some other cases, the retrieved melting layer can be too low, causing rare false positives due to rain being classified as riming. Overall, we conclude that the changes we introduced in resolution, filtering and melting layer retrieval (see Table 1) should not bias riming statistics significantly.

## 3.2 Comparison with C-band data

In Figure 7 e and f, we see the actual C-band measurements at the same time as the cloud radar measurements in Figure 7 a and b. When comparing the two scenes, we have to keep in mind the  $52 \,\mathrm{km}$  distance between the radars. As expected due to the lower sensitivity of around  $-15 \,\mathrm{dBZ}$  to  $-20 \,\mathrm{dBZ}$ , the C-band radar is lacking the cloud tops between  $7 \,\mathrm{km}$  and  $10 \,\mathrm{km}$  height. The same is true for periods with weak rain or drizzle below the melting layer. It is remarkable that in these cases, the C-band riming retrieval highlights similar time periods and regions in the cloud as the cloud radar algorithm. This supports the findings in Ockenfuß et al. (2025), that some riming events in stratiform systems can cover extended areas. Based on the duration and wind advection, they estimate a median extent of at least  $13 \,\mathrm{km}$  and around 10% of the events to extend over more than  $50 \,\mathrm{km}$  radius.




Figure 9. (a) Monthly number of riming events detected by the benchmark and operational retrieval. (b) Same as (a), but for the total monthly duration of riming.

However, these findings also imply that we cannot expect a direct correlation between the time series recorded by the two radars, given the 52 km distance separating them. This limitation is clearly illustrated in Figure 9, which aggregates data from 14 winter months between October 2021 and April 2023. On a monthly scale, we observe only a weak correlation between the two retrievals and sites. The natural variability in cloud and precipitation structure across the sites introduces considerable scatter around the 1:1 line—much more than what is seen in Figure 8.

Consequently, meaningful comparisons between the two retrievals can only be made on the basis of long-term climatological statistics. When comparing the total number of riming events detected over the 14-month period and correcting by differences in radar uptime, the C-band radar identifies 26% fewer events than the cloud radar.

From the previous experiments with the Mockup C-band dataset, we know that this discrepancy cannot be attributed to the lower temporal resolution of the C-band measurements or to any changes in the retrieval algorithm. Instead, the reduced detection rate must be inherent to the C-band system itself. Especially two properties are important: Its lower sensitivity, around  $-15 \, \mathrm{dBZ}$  to  $-20 \, \mathrm{dBZ}$ , makes it less capable of detecting riming in low-reflectivity mixed-phase clouds that occur during the cold season. In addition, the larger blind range of  $600 \, \mathrm{m}$  further limits the detectability of events close to the radar. We can quantify the relative importance from sensitivity and blind range by introducing a similar  $600 \, \mathrm{m}$  blind range to the cloud radar data. In this case, the discrepancy in event counts is reduced to 19%.

## 3.3 Spatial Distribution and Relation to Surface Precipitation

With the possibility to retrieve riming from operational radars, we are able to analyze the spatial distribution of riming probability over larger areas for the first time. Figure 10a shows the number of riming events per winter (Nov-April) for the different operational radar sites in Germany. The values are corrected for differences in the radar uptime. Generally, we see less riming

events for the northern radars compared to the ones in southern Germany. By far the most events are detected for the Feldberg (fbg) radar, located on top of the Feldberg mountain at 1494 m elevation above sealevel in southwestern Germany. In order to investigate this pattern, it is instructive to look at the precipitation distribution in Germany. For this task, we analyze the quality controlled hourly precipitation product provided by the German Weather service (DWD, 2025). The measurements are performed by PLUVIO or raineH3 tipping buckets (Saha et al., 2021; Quinlan, 2022). For every operational radar site, we average the winter precipitation accumulation from the nearest surface weather station for the years 2005 to 2025 in order to estimate the local mean wintertime precipitation. There is always a station in less than 18 km radius. For the Feldberg site, there is a surface weather station directly next to the radar at the mountain top. Figure 10b depicts the resulting mean wintertime precipitation. We can see the typical German pattern with a wet South and a dry Northeast (Kreienkamp, 2022). The Feldberg radar shows strong precipitation enhancement with 782 mm per winter, more than twice the average of 336 mm per winter for all other stations. The correlation between riming and total wintertime precipitation can be seen in Figure 11a. There is a positive correlation between the number of riming events and precipitation amount. Obviously, it is not surprising to detect more riming events at sites which are more often affected by precipitating systems. In the following, we want to investigate whether sites with greater riming and total precipitation also experience more intense precipitation, or if the increased totals are just due to longer precipitation duration. For this task, we decompose the total precipitation per site into the product of the number of hours with precipitation and the average hourly precipitation rate:

$$T = N * \hat{R}$$

$$\hat{R} = \frac{1}{N} \sum_{i}^{N} R_{i}$$

Here,  $R_i$  are the hourly precipitation rates and N is the number of hours with at least  $0.1 \,\mathrm{mm}\,\mathrm{h}^{-1}$  precipitation. N and  $\hat{R}$  are not fully independent, sites with more precipitation hours also tend to have stronger precipitation rates, as can be seen from Figure 11b. Figure 12 shows the relation to the riming events. From Figure 12a, we see that the high number of events at Neuhaus and Feldberg can mostly be explained by the greater number of precipitation hours. The number of events per hour at those two sites lies near the median of the distribution observed across all other sites. Neglecting the outliers Neuhaus and Feldberg, there is almost no correlation between number of precipitation hours and riming events for the remaining sites (grey box in Figure 12a with a pearson correlation coefficient -0.06). In Figure 12b, we see the correlation between riming events and average precipitation rate, with a focus on stronger precipitation events of at least  $1 \,\mathrm{mm}\,\mathrm{h}^{-1}$ . The correlation coefficient is 0.36 (grey box in Figure 12b, omitting the Feldberg radar).

## 3.4 Riming Onset Temperature Distribution



We define the onset temperature of a riming event as the temperature at the highest level with significant riming detections. That means, for each event, we take the temperature of the uppermost 10% of range gates, in order to get a robust estimate of the temperature level where we see the first indications of riming. The temperatures are based on radiosonde profiles. In Figure 13,

Figure 10. Spatial distribution of riming event frequency (a) and total wintertime precipitation (b) in Germany.

**Figure 11.** (a) Correlation between the number of riming events and total wintertime precipitation. (b) Correlation between the number of precipitation hours and total wintertime precipitation. Their ratio form the average precipitation rate and is shown by the colors.

**Figure 12.** (a) Correlation between the number of riming events and precipitation hours per winter. Their ratio is shown by the colors. Pearson correlation within the grey box: -0.06. (b) Correlation between the number of riming events and average precipitation rate per winter. Pearson correlation within the grey box: 0.36.

we see the onset temperature frequency distribution. The histograms are corrected for differences in radar uptime, elevation and riming frequency, as described in subsection 2.9. Despite the variability in the absolute frequency of riming events, which we already analyzed in subsection 3.3, the relative temperature distribution is very consistent between the different sites. In all cases, we see that strong riming occurs almost exclusively at temperatures warmer than  $-15\,^{\circ}$ C. This strengthens the results of Kneifel and Moisseev (2020) and Ockenfuß et al. (2025), which found similar patterns based on the analysis of four, respectively two European cloud radar sites. The notable exception here is the Feldberg radar, which has a distribution shifted towards colder temperatures. Here, we have to keep in mind that data at temperatures warmer than  $-5\,^{\circ}$ C is scarce at the Feldberg radar in winter. The slightly enhanced values at temperatures colder than  $-10\,^{\circ}$ C, compared to the other sites, may be attributed to differences in updraft speed, as further discussed in section 4.

## 4 Discussion



In subsection 3.3, we saw that for most sites, riming and total precipitation hours are uncorrelated, while there is a (weak) correlation between riming and precipitation rate. Since riming is an ice growth process and surface observations showed that rime mass can contribute the majority of surface precipitation mass (e.g. Harimaya and Sato (1989); Zhang et al. (2021) and others), such a correlation is reasonable. Nevertheless, it is difficult to observe in reality. Grazioli et al. (2015) report a

**Figure 13.** Temperature distribution of the riming event top (highest level with significant riming per event). For better visibility, the sites are separated into four subregions. The histograms are corrected for differences in radar uptime, elevation and riming frequency, as described in subsection 2.9.





300 correlation between snow accumulation and percentage of rimed precipitation based on a one month measurement campaign in the Alps. In a field campaign in Finland, Moisseev et al. (2017) also see a tendency for stronger precipitation rates in case of riming, but the effect might not be significant. Ockenfuß et al. (2025) performed a longterm analysis for a single German research site, analyzing disdrometer derived rain rates for rimed and unrimed precipitation. Based on 13 years of data, they found a statistically significant difference between the groups, with especially rain rates exceeding 1 mm h<sup>-1</sup> being more likely in the rimed group. Our analysis is probably the first one to extend over multiple years and with equidistant sites in a single climatic region.

In general, it is well known that precipitation patterns in Germany are closely linked to the orography of the region, with the highest precipitation amounts occurring in the German Alps (Kreienkamp, 2022). This geographic dependency provides a plausible framework to interpret our findings: Large scale orographic lifting enhances the production of supercooled liquid water, thereby increasing the likelihood of riming. As riming causes ice particle mass growth, this process contributes to both higher total precipitation and stronger precipitation intensities, consistent with our observations that riming and precipitation strength are climatologically correlated.

At this point, one factor that requires discussion is the influence of strong vertical air motions. While we restrict our analysis to winter months to minimize the impact of convective events, orographically induced gravity waves may still occur, especially in mountainous terrain. Because our method relies on vertical Doppler velocity measurements, there is a possibility that such wave-induced motions could be misinterpreted as signatures of riming.

Several considerations argue against this misclassification. Firstly, we apply a dedicated filter to detect and remove regions with oscillatory motions. Since the effectiveness of this filter could in principal depend on temporal resolution, we validated its performance against high-resolution measurements, as detailed in subsection 2.6. Secondly, the topography of the German sites is relatively moderate compared to the Alps. The Feldberg site, at 1494 m, is the highest in the radar network; all other radars are located below 1000 m above sea level. Moreover, if the radar installations are situated in variable terrain, the radar is often located at the highest point in the immediate surroundings. Given the additional blind range of approximately 600 m in our setup, we are generally observing well into the free atmosphere. This reduces for example the risk of persistent lee-side subsidence being misinterpreted as riming signals, as it would be the case if the radar is located at the mountain base.

Furthermore, as shown in subsection 3.4, the temperature distributions associated with riming events are remarkably consistent across all sites. This supports a microphysical origin of the observed differences rather than a dynamical one, as orographic waves would likely manifest themselves at multiple different altitudes and should not exhibit such a clear temperature dependence. The only notable exception is the Feldberg site, where the temperature distribution shows a shift toward colder values. This may indeed point to some dynamical influence at that particular location, possibly due to its higher elevation and the orientation of the black forest mountain range being orthogonal to the westerly flow.

At the same time, a link between orography and riming microphysics is also plausible for the Felberg site. In particular, updraft speeds may influence the relative temperature distribution of riming events. Both, modeling studies (e.g., Pinsky and Khain (2002)) and observational data (e.g. Snider and Brenguier (2000)) suggest that stronger updraft speeds lead to the formation of more, but smaller, supercooled liquid droplets at a given altitude compared to weaker updraft conditions. This

is due to increased supersaturation levels in stronger updrafts, which activate a larger number of cloud condensation nuclei (CCN). As these numerous CCN compete for available water vapor and are advected upward faster, the result is a population of relatively smaller droplets. Riming efficiency is strongly dependent on droplet size (DeLaFrance et al., 2024) and sharply decreases for droplets smaller than 20 µm (e.g. Lasher-Trapp (2022)). Therefore, in faster updrafts with smaller droplets, the critical point where the droplets have grown to a sufficient size for riming to happen is located higher up. This reduces the warm side of the riming histogram. In addition, stronger updrafts will prolong the residence time of snowflakes at higher altitudes, where they continue to grow due to vapour deposition. This extended growth phase may contribute to an enhancement of riming signatures at lower temperatures, thus amplifying the cold side of the temperature histogram. Taken together, these effects provide an alternative explanation for the cold-shift observed in the riming temperature distribution at the Feldberg site.

#### 5 Conclusions





In this study, we successfully adapted a riming detection algorithm—originally developed for vertically pointing cloud radars—to the birdbath scan of the operational German C-band weather radar network. The riming detection is based on the typical fall-speed of more than  $1.5\,\mathrm{m\,s^{-1}}$  of strongly rimed particles. This transfer illustrates the considerable potential of operational birdbath scans, not only for routine monitoring but also for quantitative analysis and cloud microphysical research.

Included in the method is a fast and fully automatic filtering method to remove clutter from the raw radar data. Furthermore, our riming retrieval includes an operational melting layer detection algorithm, which could serve as a valuable standalone product—offering support to forecasters or providing input for more advanced retrieval schemes.

During the adaptation process, we discovered several general challenges that arise when applying algorithms designed for research cloud radars to operational systems. In particular, differences in radar frequency bands, coarser time resolution, and the absence of auxiliary measurements or model profiles must be addressed. While we initially expected the temporal resolution of 5 min to be a major limitation, we found that the primary challenges stem from the lack of auxiliary sensor and model data, which is nowadays often used in radar based retrievals.

Our analysis of the results reveal a remarkably consistent temperature dependence of riming events across all sites, reinforcing results from previous studies that strong riming is confined to temperatures warmer than  $-15\,^{\circ}$ C. Moreover, we find a climatological link between the frequency of riming and both the total amount and intensity of precipitation, supporting a physically plausible connection between riming and enhanced precipitation rates.

Through the riming case study, we demonstrate the general scientific value of utilizing operational radar networks for atmospheric research. These systems offer equidistant spatial coverage with consistent quality monitoring and generate vast amounts of data within relatively short time periods. This enables the creation of robust long-term statistics and climatologies, and facilitates comparisons across different microclimatic regions over a larger area.

In light of these results, we believe the full potential of operational birdbath scan data remains still underutilized. Integrating these observations into established research frameworks such as Cloudnet would increase their visibility and scientific impact, providing a complement to the few but sensor-rich Cloudnet research stations. To further enhance the utility of operational




radar sites, we advocate for the installation of basic surface meteorological sensors—like temperature, pressure, and wind measurements—at all sites. In addition, the installation of a low-cost micro rain radar (?) can fill the lowermost 600 m to enable the creation of continuous profiles through the full troposphere. The live provision of such profiles to operational weather forecasters could be useful for nowcasting applications.

Finally, expanding this approach beyond Germany could yield significant benefits. Systematically recording birdbath scans from operational radar networks across Europe and providing them in a standardized format on a shared platform like Cloudnet would create a unique dataset. Such an initiative would be a major step forward for large-scale, data-driven cloud and precipitation research across the continent.

Data availability. The surface observations and radiosonde profiles used in this study are freely available via the open data server of the DWD at https://opendata.dwd.de/. For the radiosonde, we used the high resolution historical data available via https://opendata.dwd.de/climate\_environment/CDC/observations\_germany/radiosondes/high\_resolution/historical/. For the surface stations, we used the hourly historical data at https://opendata.dwd.de/climate\_environment/CDC/observations\_germany/climate/hourly/./ The Lindenberg cloud radar data is available via the Aerosol, Clouds and Trace Gases Research Infrastructure (ACTRIS) Data Centre (https://cloudnet.fmi.fi) under https://doi.org/10.60656/5b89d7d3593a4be1.

Author contributions. PO was responsible for the methodology development, data preparation, validation, visualization and writing of the manuscript. MG provided the raw radar data. MG, MF and SK gave research advice and reviewed the manuscript. SK was also responsible for funding acquisition and project administration.

385 Competing interests. MF is employed by the German Weather Service, the operator of the 17 operational radars.

Acknowledgements. This work has been supported by the DFG Priority Program SPP2115 "Fusion of Radar Polarimetry and Numerical Atmospheric Modelling Towards an Improved Understanding of Cloud and Precipitation Processes" (PROM) under grant PROM-POMODORI (Project Number 408012686). In addition, we acknowledge the German Weather Service (DWD) for providing the operational radar data, surface observations and sounding data. We thank all PIs, technicians, and instrument operators at the DWD and the Lindenberg site for ensuring high-quality, long-term measurements. We also thank ACTRIS (European Research Infrastructure for Aerosols, Clouds, and Trace gases) and the Finnish Meteorological Institute for providing Lindenberg cloud radar data. In the production of this work, tools based on artificial intelligence (AI) were used. Specifically, ChatGPT by OpenAI was used to improve the wording and formulation in some sections of the manuscript. Copilot by Github helped in code formatting and simple coding tasks like plotting. All ideas, concepts and the content of this manuscript are exclusively developed by the authors.

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
