# Peer review of "First Nationwide Analysis of Riming Using Vertical Observations from the Operational German C-Band Radar Network"

_EGUsphere, 2025_

## Referee Comment (RC1)

**Review - First Nationwide Analysis of Riming Using Vertical Observations from the Operational German C-Band Radar Network**

Paul Ockenfus, Michael Frech, Mathias Gergely, and Stefan Kneifel

**Summary & recommendation**

The manuscript describes a novel method to detect riming events using the German weather radar network. A rime fraction retrieval originally developed for Ka-band cloud radars is transferred to routinely performed vertical measurements by the C-band weather radars, so called "birdbath" scans. This allows the authors to analyze riming events nationwide in wintertime, stratiform clouds.

The proposed approach and the obtained results are sound, and fit well within the AMT scope. The paper is well structured and easy to read. I do not have major concerns, but some choices / analysis steps are not clear to me especially regarding the "mockup" that the authors use to validate the retrieval (see general comments). This should be addressed in the revision. Further, I would recommend to make the motivation (and the conclusions) less Germany-focused. This would make the manuscript more interesting for an international audience given the method can be applied/adapted to other radar networks.

**General comments**

- While the introduction does a good job introducing the readers to the content of the manuscript, I am missing the "bigger picture motivation". There is also a strong focus on Germany. This makes the study less accessible/interesting for international readers. I would suggest to start with a more general paragraph on why riming is important and why large-scale monitoring is crucial to study spatial variability. And afterwards go into your dataset (German radar network), but mention that this could be applied to other radar networks.

- I don't understand certain choices regarding your mockup retrieval. To act as a validation for your method, the mockup should be as close as possible to the C-band data. However you only resample the Ka-band cloud radar data to the C-band spatial and temporal resolution if I understood correctly. The blind range of 600 m is not included in the mockup and neither the lower. I am wondering why as I would assume both could be implemented rather easily and make the validation more sound. If you want to show the contribution of the lower temporal resolution vs. sensitivity and blind range, why not include two mock-up version? Then you can show the discussion in L265-271 also visually, e.g. by

including both mockups in Fig. 7 as an example. In addition, I would argue calling the resamples Ka-band data "C-band mockup" is misleading as readers might expect you performed some kind of frequency correction.

- References are missing at several location (seems to be a bibtex issue since there are (?) instead). Please make sure to double check all references. I will list all occurrences that I notices in technical corrections.

**Specific comments**

- L55-60 (there is an issue, these are more than 5 lines): How is the separation of rain and ice part done? Do you mean liquid and ice instead of rain and ice? Is this done via cloudnet? What about mixed-phase regions, which are important especially in the context of riming? Please include more details and/or a reference and discuss uncertainties of the used approach.
- L55-60: I don't understand why you stress "unique" relation. What do you mean by unique?
- Figure 1: the yellow x might be hardly visible if this plot is smaller in the final paper. I suggest to make the symbols all a bit larger. Also please mention in the figure description that the station abbreviations are included in the brackets. This might be missed at first glance, which is confusing later.
- L96: "liquid layers in ice clouds" is an oxymoron, no? → Write mixed-phase clouds or ice-containing clouds
- L99-101: But ERA5 is available for the station locations, right? Could DWD automate the download and store alongside the radar data? I'm asking, because right now it sounds like the problem is with ERA5. But I assume you want to stress that Cloudnet like data storage would be ideal? Maybe rephrase this section to get your point across better.
- L111: So, are you applying relations developed for X-band to the C-band data (since everything is in the Rayleigh regime for both)? If yes, please state this explicitly.
- L124: I understand that you need height dependent thresholds, but I have some difficulty with the site dependence. Won't that impact your statistics? Especially if there were any offsets between sites? How different would your results be if you use universal height dependent thresholds for all sites? Based on which timespans are the thresholds defined? Or is there a threshold for each birdbath scan? Please include a few more details and a discussion how much the choices in setting the thresholds impacts your results.
- L134: how many riming events do you miss due to the 600 m minimum? I suggest to use your mock-up with and without a 600 minimum to quantify.

- L139: What about mixed-phase regions of the cloud? Do you mean ice-containing?
- L145: Your riming detection only works for FR>0.6 if I am not mistaken. I would therefore argue that you can't say riming is rare, only that strong riming cases are rare.
- L155: "from manual analysis" I'm inclined to trust you, but this might be nice to include in an appendix to ensure reproducibility.
- L162: Can it happen that a riming event is detected as a melting layer? If riming occurs, I would also expect an increase in v, faster falling particles below  and slower falling particles above the event.
- L165: You write that this makes it less likely that riming is misinterpreted as the melting layer, but what about larger-scale riming events?
- Figure 3: Looking at this I'm wondering. What if you have a melting layer and re-freezing below (probably very rare, but might occur)?
- Figure 3: This shows one case / site. What about a comparison with more data? For example, you could compare this detection using the mock-up data to the Cloudnet melting layer detection?
- L191: for what amount of data did you do the mentioned systematic tests?
- Figure 4c. I suggest to make this subfigure more intuitive to read by using different colors. For example, you could use similar colors for the true positives & the false negatives and similar colors for the false positives & true negatives. And maybe label this also accordingly, so that it's clear at first glance which are the "correctly identified" pixel and which are "wrong".
- L213-214: I don't understand what you mean with this sentence. Do you mean by the 2 min km filtering you remove 17 riming cases?
- Figure 6: Mention that this shows example sites in the figure description. Also please state clearly in the text why you focus on winter. Possibly where you introduce the convection filtering.
- Figure 8: I recommend to include RMSE.
- L241: different results for some months → is this random or is there some seasonality?
- L250: why are you only using the Lindenberg Ka band to compare. Isn't there also a cloud radar in Munich that is close to the Isen DWD radar? I just quickly looked at the cloudnet stations in Germany, so I could be mistaken. If there are other "close-by" cloud radars, I think it would be highly valuable to do the comparison for a second site.
- L272: You only detect riming for stratiform situation if I understood correctly. Do you still compare to all surface precipitation? How much precipitation from convective cases do you have in winter for your dataset? Please discuss. Also, does this include rain and snow?

- Figure 10: what do the different circle sizes mean? Please include a legend
- Figure 13: Please include the amount of data per temperature bin in theses figures to give context how reliable the data is. Also, it would be great to include a measure / statistic test if the distributions are significantly different for different subregions or stations or not.
- L311: But you only showed a weak correlation, right? Please include this here.
- L313: this should be stressed earlier in the text. Also, please include for example percentages how large (or rather small) the contribution of convective data in winter is compared to stratiform.
- L346: I recommend to remind the reader what a birdbath scan is. This will make the text more accessible for people only reading the conclusions.
- L374: Include a reference for this statement (I'm assuming the Kneifel & Moisseev paper or whoever they cite).
- L365-366: what is hindering the integration to Cloudnet? Are you in contact with ACTRIS and DWD?
- L373: Why the focus on Europe? Could the method be applied to international radar networks?
- Data availability: Will you provide the code? Maybe a python tool? This looks like a nice retrieval that could be applied or adapted to radar networks in other countries.

**Technical corrections**

- L71: reference missing
- L91 (and later): remove the brackets around the year in the reference → citep instead of (citet).
- L140: reference missing
- L201: Figure 5b is mentioned before Figure 5a → please flip
- Figure 4: please use color-blind friendly color maps.
- L236 (and likely earlier): Usually, Figure is written out at the beginning of a sentence and abbreviated to Fig. otherwise.
- L 270-275 (again way more lines than 5): the equations: I don't think all parameters are described.
- L369: reference missing

---

## Referee Comment (RC2)

**Paul Ockenfuß et al.: "First Nationwide Analysis of Riming Using Vertical Observationsfrom the Operational German C-Band Radar Network"**

**Overall assessment**
The authors use the vertical (zenith) "birdbath" scans of the 17-station German operational polarimetric C-band radar network (operated by Deutscher Wetterdienst, DWD). The birdbath scan is typically used for calibration, but here is used to retrieve microphysical information (riming in mixed-phase clouds).
Taking operational C-band birdbath scans and using them for microphysical retrievals is a novel and elegant reuse of existing infrastructure.
This is a highly relevant topic, and the manuscript is generally well written, in most parts easily understandable, and I highly recommend publication of the paper after addressing the comments below.
Figures are generally clear and well designed. A few adjustments to captions and colormaps could further improve readability and accessibility.

**General comments**
- The introduction would benefit from a stronger motivation for why detecting riming and understanding its drivers is important for cloud physics, precipitation processes, and numerical weather prediction.
- use either the "German Meteorological Service" or "German Weather Service" consistently throughout the manuscript
- as the other referee notes, your perspective is very Germany-centered and only in the last paragraph do you expand your view to the broader (European) perspective. *Which other national meteorological services operate similar C-band networks? Do any routinely perform birdbath scans?* Clarifying this would help position the work within a broader operational context.
- The original Kneifel & Moisseev (2020) retrieval used an average of MDV over a relatively long (several minutes) window to get rid of turbulence effects and wavy structures. How does this scale with the 15 rays every 5 minutes data from the C-Band radar network? Maybe I missed it in the text, but I think there should be a brief description of the Kneifel & Moisseev (2020) retrieval, how it was improved in Ockenfuß et al. (2025) and how it is finally applied to the C-Band data. You could list (maybe in the table) how many MDV values are averaged for each of the radar data sets, if you use a boxcar filter, and which MDV-FR relation from Kneifel & Moisseev (2020) is used for which of the data sets.

**Specific comments**
- line 19f: Clarify whether research cloud radars *scan* in the vertical or whether they typically perform vertically pointing observations without scanning. I thnk that "vertically pointing observations" (and not scans) are the standard measurement strategy.
- what should be Eq. (1), the number is missing here for some reason: I think that the subscript "c" for "crystal" is potentially misleading, as "crystals" typically denote pristine individual ice particles. Since the retrieval mainly targets riming of snowflakes, consider using notation referring to "unrimed snow," e.g. *us* (as in Moisseev et al., 2017), or something similar refering to both unrimed aggregates and crystals.
- l. 62: The first figure referenced is Fig. 7a,b, which disrupts the flow. Consider omitting the figure reference here and describing the retrieval conceptually, returning to the examples later.
- Figs. 3 and 4 and 7: use the same colormap for the Doppler velocity. I generally like the use of colorblind friendly color schemes as in Fig. 3. In this context, please also ensure Fig. 1 is accessible to readers with color-vision deficiencies.

- l. 62: Fig. 7a and b : Please provide site and instrument information for the frontal systems shown in the referenced figures in the text to improve understandability.
- l. 71: there is a reference missing.
- l. 74f: "In the vertical, the intrinsic data resolution is 60m, but sampling is performed at 25m resolution (Gergely et al., 2022)." Please clarify how a dataset with intrinsic 60 m resolution is sampled at 25 m, are spectra interpolated or is the FFT performed on overlapping parts of the time series? And does it make sense to look at the data in 25 m resolution then?
- l. 92 : I doubt a bit that it is possible to apply the EDR retrieval by Borque et al. (2016) to the 5min resolution birdbath scan, even if you took relatively long (e.g. 20 minute) windows to apply the retrieval. I think it would still be too few data points to fit turbulence spectrum slopes reliably. Please check this, or remove the reference here.
- l. 93 "equipped"
- l. 102ff: Why are you addressing the points 1-3 in reverse order, i.e. start with the last point and then moving on to the second and finally the first? Consider changing the order and separating the considerations regarding each of the three points with a line break to improve readability.

**Mockup data**
- 105ff: how are you upsampling the 35 m resolution MDV data to 25 m? Please state which kind of interpolation you are using in the text.
- You are not taking every 10[th] cloud radar data profile but every 10[th] cloudnet profile, correct? I understand that cloudnet categorize data are already 30 s averages of the raw MDV:
https://github.com/actris-cloudnet/cloudnetpy/blob/main/cloudnetpy/categorize/radar.py#L347
I guess this has implications for the mockup data and all the error considerations downstream (possibly the kappa filter): If the mockup data are 30 second averages, you miss small-scale variability that you would have in unaveraged MIRA-35 data. The goal is to use data similar as possible to the C-Band observations.
I strongly recommend having another close look at the mockup data and making sure that it is really comparable to the C-Band birdbath observations. My guess is that the number of individual samples, coherent and uncoherent averages, going into one time-height observation in the C-Band radar data should be as close as possible to the number of samples in one time-height observation of the mockup data. Are the C-Band data averaged over 15 s more similar to the 30 s cloudnet resolution or to the "raw" MIRA-35 observations? Or does it make sense to aggregate a couple of raw MIRA measurement time steps to get to the resolution of the C-Band 15-s-data? My guess would be that 30 seconds of MIRA observations contain more than your 15*1024 pulses. It might make sense to actually dig into the spectra and (visually) make sure that the data look similar.
- For creating the mockup data, why didn't you take into account the lower sensitivity of the C-Band radar and the blind range, i.e. artificially worsen the Ka-Band data? I think it wouldn't be a big deal to exclude pixels below the height-dependent Ze threshold, or, for selected cases such as in Fig. 7, even go into the Doppler spectra and cut them at a higher threshold and then recompute MDV to see what happens when you miss all the liquid water peaks.

**Some more specific comments**
- l. 110ff: I do not fully understand the meaning of these sentences. Please clarify: Are you saying that you are using the fit coefficients by Kneifel and Moisseev (2020) derived for X-Band on your C-Band data (but probably the ones for Ka-band on the mockup data?) and that X-Band and C-Band are sufficiently close together, the transition size where Mie effects play a role in X-Band would be 1 cm, and this can be neglected except for very large aggregates? You could add the information which FR-MDV relation is used, to Table 1.
- l. 131 "for most of the height range"
- l. 140: missing reference.

**Melting layer detection**
The Cloudnet melting layer detection uses a combination of a peak in LDR (is this available for your cloudnet data?) and the criterion of MDV above 1 m/s at melting layer base, and, as a fallback if no LDR is available, only profiles of MDV, and, if available, spectrum width. This should be stated somewhere in the text.
Why develop a new algorithm instead of using or extending the Cloudnet MDV/ width one? In general though, it looks like the melting layer detection method proposed here seems to work very well, and this might also be of interest for Cloudnet developers. This could motivate a small comparison study, like in Fig. 3.
The cloudnet melting layer code detection is here:
https://github.com/actris-cloudnet/cloudnetpy/blob/main/cloudnetpy/categorize/melting.py#L128

**More specific comments**
- l. 157: fallspeeds (lower case) and "criterion" instead of "criteria"
- l. 174: "In Fig. 2c, only winter months (Nov to Apr)… ": this sentence is somewhat out of context here. I would mention this earlier, when discussing Fig. 2 (without having explained the melting layer detection yet but referring the reader to the next section), and maybe also write it in the caption of Fig. 2.
- Side question: Why consider only winter months? You are mentioning it later in the Discussion, that it is to exclude convection, but I think this should be explained much earlier.
- l. 176f: "Only if no reliable detection was possible, we extrapolate the last value up to 1 h and otherwise use the value from the nearest sounding station as a proxy. In the latter case, at most 12 h are tolerated between sounding and radar profile." How far are the nearest sounding stations for the different radars?
- l. 186: replace the "original cloud radar resolution" with the cloudnet categorize/ classification time resolution or something similar
- approx. l. 200 (line numbering is broken there): collocated, not colocated
- Fig. 5a and b are referenced in opposite order in the text, consider switching them
- l. 204: 20,000 instead of 20.000
- l. 212 remove duplicate "percent"
- l. 222ff: Your point 3 is scientifically meaningful and worth further discussion.
- Fig. 6: Maybe I missed it somewhere in the text, but I am missing a definition of "winter" and a discussion why these months were chosen and not differently. Sometimes "winter" refers to the months of DJF only, but it is different in your case. In one place, you write November to April, in another "14 months between October 2021 and April 2023".
- Also Fig. 6: Even though it's kind of obvious, I would also mention why there is a vertical dashed line at 0 degrees (no riming at higher temperatures)
- Fig. 7: This is a matter of taste, but I think it would benefit from adding the site and date as column headers.

- Section 3.3:
- beginning of Section 3.3 (the line numbers are broken again): I would mention the second outlier, Neuhaus, much earlier in the text. You mention that the Feldberg site is influenced by orography. Any ideas why the Neuhaus site sticks out? Interestingly, it does not stick out with respect to temperature distribution of the riming onset in Fig. 13. Any ideas why?
- last 2 lines before the equation: For better readability, I would refer to the letters in the equation in the explanatory text, i.e. "we decompose the total precipitation per site, T, into the product of the number of hours with precipitation, N and the average hourly precipitation rate $\hat{R}$:

- l. 272ff: Equation numbers are missing again.

- Fig. 10: In 10b, it's not so easy to see the wet south and dry northeast. Consider changing the colorbar a bit
- l 273ff: The discussion of Figs. 11 and 12 could use a bit more detail. I think these are some of the main results in this paper and I find it a bit hard to follow. In particular Fig. 12b is only discussed very briefly.
- l 290: "who" instead of "which"

**Fig. 13 and vertical air motions**
- l. 316ff: vertical air motions: The kappa filter removes updrafts, but may leave stationary or slowly varying downdrafts unflagged. Stationary downdrafts could be misinterpreted as riming by the algorithm. I guess that those downdrafts however would not shift the entire temperature distribution in Fig. 13 to lower temperatures for the Feldberg site, as you observe, but might increase riming frequency equally across most of the temperature range (if we assume that the downdrafts influence the entire atmospheric column similarly). In connection with the lower number of observations at higher temperatures due to higher elevation of the Feldberg site, it's a bit hard to wrap the head around the different effects different factors could have on the temperature distribuion curve.
While for the other (not orography influenced sites) I would expect a probability of 0 at very low temperatures, say below -20°C, at the Feldberg site, there could be also riming at temperatures below that range, maybe even below -40°C, due to misclassifications, e.g. if the entire atmospheric column is in a downdraft for several minutes. It is somewhat eye-catching that fbg is the only curve extending down to the x-axis lower limit, and maybe even beyond.
*How does the curve look like for lower temperatures below the x-axis lower limit in Fig. 13?* This can give a hint on the uncertainty of the method introduced by orography.
*- If you are not normalizing to an area of 1 below the curve but rather look at the likelihood of riming per temperature bin as Kneifel & Moisseev (2020) did in their Fig. 8, how does fbg then compare to the other sites?*
- l. 331ff: the idea that stronger updrafts might be responsible for the shift in the temperature distribution towards colder temperatures at first sounds plausible, but aren't these the cases that the kappa filter flags as convective/ updraft?

**More general comments:**
- did you look into whether there is a correlation of FR and precipitation intensity?
- Conclusions: One point from earlier in the manuscript could be brought up again here: Saving high-resolution model profiles of T, p, q and wind for all the weather radar sites should not be a big deal as they are also being saved for Cloudnet sites, correct?
- In general, the Conclusions are beautifully written, and motivating.

---

## Author Comment (AC1)

**Revision 1**

Paul Ockenfuß, Michael Frech, Mathias Gergely, Stefan Kneifel

December 30, 2025

**Preface**

We thank both reviewers for taking the time and reviewing our manuscript. Those are exceptionally valuable and in-depth reviews. We think all of the raised points are valid comments and we tried to respond to each point as good as possible.

**Structure**

This document contains up to three items for each review comment. **Comment** contains the original comment by the reviewer. **Response** is our response to the comment, explaining our thoughts and decisions. **Changes to manuscript** optionally lists the most important adaptations to the manuscript with regard to this comment. Generally, line numbers refer to the submitted version of the manuscript. For a full list of changes, please also have a look at the provided differential document, showing all differences between the submitted and revised version of the manuscript.

**Reviewer 1**

**General Comments**

R1.1 **Comment:** While the introduction does a good job introducing the readers to the content of the manuscript, I am missing the "bigger picture motivation". There is also a strong focus on Germany. This makes the study less accessible/interesting for international readers. I would suggest to start with a more general paragraph on why riming is important and why large-scale monitoring is crucial to study spatial variability. And afterwards go into your dataset (German radar network), but mention that this could be applied to other radar networks.

**Response:** We restructured the introduction as you suggested. We moved the riming introduction to the beginning and try to highlight that the German Weather radars are just an example. We focus on Germany, because this is, to our knowledge, the only network that already explores the potential of the birdbath scan. Technically, as you say, this could be applied to many other (European) radar networks.

R1.2 **Comment:** I don't understand certain choices regarding your mockup retrieval. To act as a validation for your method, the mockup should be as close as possible to the C-band data. However you only resample the Ka-band cloud radar data to the C-band spatial and temporal resolution if I understood correctly. The blind range of 600 m is not included in the mockup and neither the lower. I am wondering why as I would assume both could be implemented

rather easily and make the validation more sound. If you want to show the contribution of the lower temporal resolution vs. sensitivity and blind range, why not include two mock-up version? Then you can show the discussion in L265-271 also visually, e.g. by including both mockups in Fig. 7 as an example. In addition, I would argue calling the resamples Ka-band data "C-band mockup" is misleading as readers might expect you performed some kind of frequency correction.

**Response:** We agree that the term "mockup" is misleading. We renamed it to "Ka-band resampled". This might already resolve the confusion: The mockup/resampled dataset is not meant to resemble the C-band as close as possible in any regard. As described in Sec. 2.3, there are several typical key differences between research cloud radars and operational radars: Frequency, time resolution and additional sensor information. As stated in line 104, the main motivation for the resampled dataset was to asses the effect of time resolution: *"To quantify the influence of point two, the difference in resolution, we create a "mockup" C-band radar dataset from cloud radar data."* This resampled dataset was used to validate and adjust the filters like the 20 min boxcar-convection filter for the coarser time resolution. Other aspects, like the blind range and sensitivity, are deliberately kept similar to the benchmark retrieval as it is published in Kneifel 2020, and Ockenfuss, 2025. Otherwise, we would not be able to tell if, for example, less riming events are detected because the coarser time resolution does not permit so, or because the blind range is simply different from the benchmark. As you note correctly, at this point, we could create several further mockup datasets, each with different combinations (e.g. fine/coarse time resolution, low/high blind range, etc.). We decided to focus on the given combination (time resolution yes, blind range and sensitivity no), since it is the most relevant for retrieval development. While we can try to adapt techniques and filters like the convection filter or the melting layer detection to the low time resolution, there is nothing we can do to mitigate blind range and sensitivity effects: Low reflectivity regions and the near-surface region are simply not observable with the C-band. However, you are right that we can later on introduce these points just to diagnose the effect. For this reason, in line 266 onward of the initial manuscript, we discuss the effect of removing all events from the resampled retrieval results, which would be significantly affected by the blind range of the real C-band radars. In the revised manuscript, we additionally discuss the effect of removing events which are significantly affected by the reduced sensitivity. To do this, we introduce a -17 dBZ threshold to the cloud radar data, consistent with the average sensitivity of the 'Essen (ess)' and 'Prötzel (pro)' radars (compare Figure 3). We found that the sensitivity limit does not have a strong effect: No significant reduction in detections is observed for the Lindenberg data and only three events (1%) are discarded for the Jülich data. Please note that for a perfect replication of the C-band, just applying the sensitivity of the C-band to the Ka-band data is not enough, since the Ka-band scattering transitions into the non-Rayleigh regime for large particles. This should partly be accounted for, as discussed in Sec. 2.3 line 109 onward, by the difference in the FR-MDV relation between Ka- and C-band (X-band). However, a perfect "mockup" replication of actual C-band measurements would probably require a numerical weather model including riming, coupled with a radar forward model. This is beyond the scope of this manuscript. We also included the blind range and sensitivity in Fig. 7. For this case, it does make almost no difference (see the image below), since the melting layer is above the C-band blind range and the low-reflectivity regions of the cloud are usually not the areas with riming.

**Changes to the Manuscript:** We renamed "mockup" to "Ka-band resampled", to better capture the intent and definition of this dataset.

[Figure]

Figure 1: Example cases of Apr. 14 & 28, 2023. The blind range and sensitivity of the C-band radars is applied to the resampled Ka-band dataset.

R1.3 **Comment:** References are missing at several location (seems to be a bibtex issue since there are (?) instead). Please make sure to double check all references. I will list all occurrences that I notices in technical corrections.

**Response:** Yes, this was likely a bibtex/autoref issue. We hopefully corrected them all.

**Specific Comments**

R1.4 **Comment:** L55-60 (there is an issue, these are more than 5 lines): How is the separation of rain and ice part done? Do you mean liquid and ice instead of rain and ice? Is this done via cloudnet? What about mixed-phase regions, which are important especially in the context of riming? Please include more details and/or a reference and discuss uncertainties of the used approach.

**Response:** This is essentially a separation between the warm, non-ice-containing part of the cloud at temperatures above 0°C and the cold, ice containing part below 0°C. The details are described in Ockenfuß et al. [2025]. We added a clarification that this is done based on the Cloudnet target classification [Hogan and Connor, 2004] and the cloudnet model temperature profiles.

R1.5 **Comment:** L55-60: I don't understand why you stress "unique" relation. What do you mean by unique?

**Response:** Monotonic would be the correct word here. We changed it accordingly in the manuscript.

R1.6 **Comment:** Figure 1: the yellow x might be hardly visible if this plot is smaller in the final paper. I suggest to make the symbols all a bit larger. Also please mention in the figure description that the station abbreviations are included in the brackets. This might be missed at first glance, which is confusing later.

**Response:** Changed accordingly.

R1.7 **Comment:** L96: "liquid layers in ice clouds" is an oxymoron, no? Write mixed-phase clouds or ice-containing clouds

**Response:** We rephrased it as "to detect liquid layers in mixed-phase clouds".

R1.8 **Comment:** L99-101: But ERA5 is available for the station locations, right? Could DWD automate the download and store alongside the radar data? I'm asking, because right now it sounds like the problem is with ERA5. But I assume you want to stress that Cloudnet like data storage would be ideal? Maybe rephrase this section to get your point across better.

**Response:** Yes, the data is technically available in the tape archive of the ECMWF. But, as you state correctly, we were not able to download it. About two years ago, we discussed this issue with the ECMWF data support, but found no easy solution. The problem seems to be that, in order to access a profile over a single point in space and time, the tape with the full global data for this specific time needs to be retrieved from the tape storage system. Therefore, in order to create a multi-year timeseries over a single point, basically all tapes in the storage need to be accessed at least once. This was, at least two years ago, beyond the capabilities of the ERA5 online access quota. To our knowledge, Cloudnet solves this problem by continuously downloading the data since several years, which was obviously not an option for us.

R1.9 **Comment:** L111: So, are you applying relations developed for X-band to the C-band data (since everything is in the Rayleigh regime for both)? If yes, please state this explicitly.

**Response:** Correct, we changed the sentence accordingly.

R1.10 **Comment:** L124: I understand that you need height dependent thresholds, but I have some difficulty with the site dependence. Won't that impact your statistics? Especially if there were any offsets between sites? How different would your results be if you use universal height dependent thresholds for all sites? Based on which timespans are the thresholds defined? Or is there a threshold for each birdbath scan? Please include a few more details and a discussion how much the choices in setting the thresholds impacts your results.

**Response:** This is a valid point. In general, the thresholds are site dependent, because the specific clutter lines appear at different heights between the sites. However, apart from the clutter lines, the overall shape is similar between the radars. In Fig. 2c in the manuscript, we originally decided to show only the Essen radar site as an example. We suggest including the following Figure 2 in the appendix of the manuscript, to show the reflectivity histograms for all other radars, including the threshold lines to separate the clutter from the signal. To determine whether the variability in the thresholds between the sites has an influence on the riming statistics, we focus on the 0 km to 5 km height region, which is most relevant for riming. In this region, the average sensitivity of the radars is $-14.2$ dBZ $\pm 2.2$ dBZ (note that, for comparison, a cloud radar has an average sensitivity of around -45 dBZ in this height region). We then investigated whether this variability explains some of the variability in riming event frequency, as we see it in Fig. 10a in the manuscript. However, we found no correlation (pearson coefficient 0.05) between the sensitivity variability and riming frequency, as shown in Figure 3 below.

**Changes to the Manuscript:** We added Figure 2 to the appendix of the manuscript. For brevity, the results of Figure 3 are summarized as text in the discussion section of the manuscript.

R1.11 **Comment:** L134: how many riming events do you miss due to the 600 m minimum? I suggest to use your mock-up with and without a 600 minimum to quantify.

**Response:** See the discussion in line L270 of the original manuscript.

R1.12 **Comment:** L139: What about mixed-phase regions of the cloud? Do you mean ice-containing?

**Response:** Yes, ice-containing would be a more appropriate choice.

R1.13 **Comment:** L145: Your riming detection only works for $FR > 0.6$ if I am not mistaken. I would therefore argue that you can't say riming is rare, only that strong riming cases are rare.

**Response:** Corrected!

R1.14 **Comment:** L155: "from manual analysis" I'm inclined to trust you, but this might be nice to include in an appendix to ensure reproducibility.

**Response:** We created Figure 4 and added it to the appendix. Note that our value is almost identical to the value of 0.02 used by Wolfensberger et al. [2015], if we account for the 3x25 m vertical range and the normalization factor of 8 for the Sobel matrix. We also corrected the unit, which must be $m\ s^{-1}$.

[Figure]

Figure 2: Same as Fig. 2c, but for all of the 17 radar sites. The blind range is shown as grey shading.

[Figure]

Figure 3: Frequency of riming events, as a function of the average sensitivity in the 0 to 5 km height region, per radar.

[Figure]

Figure 4: Velocity gradient, calculated by 3x3 Sobel filter as in Wolfensberger et al. [2015]

R1.15 **Comment:** L162: Can it happen that a riming event is detected as a melting layer? If riming occurs, I would also expect an increase in v, faster falling particles below and slower falling particles above the event.

**Response:** Yes, theoretically this could happen. However, as demonstrated in Figure 4, the gradient created by the melting layer is usually much more distinct than the gradient resulting from riming.

R1.16 **Comment:** L165: You write that this makes it less likely that riming is misinterpreted as the melting layer, but what about larger-scale riming events?

**Response:** Usually, as mentioned in item R1.15, even in the presence of large-scale riming, the melting layer has a stronger gradient in velocity. The second criteria is mostly relevant for very small-scale features (single pixels, probably often noise or noisy velocity detections in areas with low reflectivity like near cloud top), which produce strong, local gradients.

R1.17 **Comment:** Figure 3: Looking at this I'm wondering. What if you have a melting layer and re-freezing below (probably very rare, but might occur)?

**Response:** As you say, this is really rare and therefore statistically not relevant. Currently, our algorithm will always choose the level with the biggest gradient and velocity difference. In the case of a double melting layer, we expect this to be the higher level, since there the shape changes from (rimed) aggregates or crystals to droplets. If droplets refreeze, the resulting ice particles will still have a very high effective density (almost solid ice), so we expect the velocity transition for the second melting to be weaker than the first transition. However, we have never identified such a case. If found, this might be an interesting case for a separate study.

R1.18 **Comment:** Figure 3: This shows one case / site. What about a comparison with more data? For example, you could compare this detection using the mock-up data to the Cloudnet melting layer detection?

**Response:** Fig. 3c shows a statistical evaluation of many cases. Instead of the Cloudnet melting layer, we decided to compare with radiosonde data. The radiosonde, if available, provides a direct in-situ observation of the temperature profile, which we preferred over a retrieval-retrieval comparison. For the Essen site, the radiosonde launch site is perfectly collocated.

R1.19 **Comment:** L191: for what amount of data did you do the mentioned systematic tests?

**Response:** We tested several days in November and December 2022, but ultimately took the more convective case of 04. July, 2021, shown in Fig. 4 in the manuscript, as a benchmark.

R1.20 **Comment:** Figure 4c. I suggest to make this subfigure more intuitive to read by using different colors. For example, you could use similar colors for the true positives & the false negatives and similar colors for the false positives & true negatives. And maybe label this also accordingly, so that it's clear at first glance which are the "correctly identified" pixel and which are "wrong".

**Response:** We included the labels and rearranged the colors, such that the true positive and true negative have the green/blue color, while the false negative and positive have the orange/pink color. The colors itself are chosen to be colorblind friendly. We tried to create a compromise between similarity of the colors in the true/false groups and colorblind friendliness.

R1.21 **Comment:** L213-214: I don't understand what you mean with this sentence. Do you mean by the 2 min km filtering you remove 17 riming cases?

**Response:** No, this means we remove events consisting of less than 17 samples or "pixels" in the time-height view (1 pixel = 5 min x 25 m). This threshold is consistent with Kneifel and Moisseev [2020] and Ockenfuß et al. [2025]. The reason is that there are always some single, noisy pixels with enhanced velocity, as also mentioned in item R1.16. We do not want to interpret them as riming. We clarified that in the manuscript.

R1.22 **Comment:** Figure 6: Mention that this shows example sites in the figure description. Also please state clearly in the text why you focus on winter. Possibly where you introduce the convection filtering.

**Response:** These radars were selected because they serve well to illustrate the different factors that influence the number of observations. We focus on winter, since, even though we implemented two convection filters based on variability of the Doppler velocity and maximum reflectivity, we want to be as conservative as possible for the precipitation and temperature analysis. Therefore, we focus on the winter months, where deep convection is rather rare in Germany. We clarified this in the manuscript.

R1.23 **Comment:** Figure 8: I recommend to include RMSE.

**Response:** We included the RMSE and the r2 value in the plots.

R1.24 **Comment:** L250: why are you only using the Lindenberg Ka band to compare. Isn't there also a cloud radar in Munich that is close to the Isen DWD radar? I just quickly looked at the cloudnet stations in Germany, so I could be mistaken. If there are other "close-by" cloud radars, I think it would be highly valuable to do the comparison for a second site.

**Response:** Yes, it is true that e.g. in Munich, there is also a Ka band radar, even maintained by some of the authors of this manuscript! Nevertheless, we decided to focus on the DWD Lindenberg radar for this analysis, since it has one of the longest vertically looking time series in Germany. The radar at the University of Munich was in the past more frequently involved in specific measurement campaigns and therefore running special scan patterns or radar settings. In Ockenfuß et al. [2025], there is a comparison between the cloud radars in Lindenberg and Jülich. In the development of the C-band algorithms, we also did the analysis for the cloud radar Jülich in combination with the operational radar Essen, in addition to Lindenberg and Prötzel. This is now shown in Figure 5 and Figure 6. Since those plots qualitatively show the same results as the Lindenberg plots, for brevity we focused on Lindenberg in the original manuscript (the Lindenberg-Prötzel radars are also located closer than the Jülich-Essen radars). We now added Jülich to Fig. 8 and Fig. 9 of the manuscript.

R1.25 **Comment:** L272: You only detect riming for stratiform situation if I understood correctly. Do you still compare to all surface precipitation? How much precipitation from convective cases do you have in winter for your dataset? Please discuss. Also, does this include rain and snow?

**Response:** Yes, we compare to all wintertime surface precipitation. This is one of the reasons we restrict the analysis to wintertime precipitation. As also mentioned in item R1.29, we added the amount of heavily convective cases, detected by the reflectivity based filter, in the methods section. Removing them from the surface precipitation time series is difficult, since the gauges are not perfectly collocated with the radars. This dislocation is the reason we compare the gauges and radars on a climatological level. Note that, in agreement with our

[Figure]

Figure 5: (a) Monthly number of riming events detected by the Lindenberg benchmark and resampled retrieval. (b) Same as (a), but for the total monthly duration of riming. (c) Same as (a), but for Jülich and Essen. (d) Same as (b), but for Jülich and Essen.

[Figure]

Figure 6: (a) Monthly number of riming events detected by the Lindenberg benchmark and Prötzel operational retrieval. (b) Same as (a), but for the total monthly duration of riming. (c) Same as (a), but for Jülich and Essen. (d) Same as (b), but for Jülich and Essen. The different number of samples between (a,b) and (c,d) is due to differences in radar uptime.

findings, wintertime thunderstorms are very rare in Germany (see, for example, Fig. 11 in Taszarek et al. [2019]).

R1.26 **Comment:** Figure 10: what do the different circle sizes mean? Please include a legend

**Response:** The circle sizes show the same values than the colors. We included them as overlay in the colorbar.

R1.27 **Comment:** Figure 13: Please include the amount of data per temperature bin in theses figures to give context how reliable the data is. Also, it would be great to include a measure / statistic test if the distributions are significantly different for different subregions or stations or not.

**Response:** We added a corresponding figure in the Appendix. We also added a statistical analysis. However, this turned out to be more complicated than one would expect. A simple statistical test is not enough, since we want to check whether the riming probability as a function of temperature is shifted, while acknowledging the fact that the precipitation climatology creates absolute differences in the riming counts. This requires a model for the riming probability - temperature relation. We formulated such a model and then analyze the statistical significance of the factor governing the mean temperature per site.

**Changes to the Manuscript:** Please read the new section about the statistical temperature analysis, added to the appendix.

R1.28 **Comment:** L311: But you only showed a weak correlation, right? Please include this here.

**Response:** Included.

R1.29 **Comment:** L313: this should be stressed earlier in the text. Also, please include for example percentages how large (or rather small) the contribution of convective data in winter is compared to stratiform.

**Response:** We included the amount of convective cases in the Methods section, when discussing the implementation of the filters.

R1.30 **Comment:** L346: I recommend to remind the reader what a birdbath scan is. This will make the text more accessible for people only reading the conclusions.

**Response:** Added! ("vertically looking 'birdbath' scan)

R1.31 **Comment:** L374: Include a reference for this statement (I'm assuming the Kneifel & Moisseev paper or whoever they cite).

**Response:** Which exact statement are you referring to in line 374?

R1.32 **Comment:** L365-366: what is hindering the integration to Cloudnet? Are you in contact with ACTRIS and DWD?

**Response:** Yes, we are in contact with both. Currently, the operational birdbath scan is not yet publicly released by the DWD, but this is planned. Once available in the DWD OpenData portal, an integration into other databases would be feasible.

R1.33 **Comment:** L373: Why the focus on Europe? Could the method be applied to international radar networks?

**Response:** Very likely yes, as long as the birdbath scan is available! Unfortunately, the Nexrad radars are, to our knowledge, mechanically not capable of pointing vertically. In Europe, the

majority of the 217 radars within the OPERA European radar network are doing birdbath scans. Currently, we do not have exact numbers, but OPERA is now planning to collect this information. We do not know how the situation is for networks outisde of Europe and the US, but we do hope to trigger further activity into this direction with this manuscript.

R1.34 **Comment:** Data availability: Will you provide the code? Maybe a python tool? This looks like a nice retrieval that could be applied or adapted to radar networks in other countries.

**Response:** We are working on this. We are already in contact with Cloudnet to publish and implement the "reference" retrieval into the Cloudnet data portal as experimental product. We now also uploaded the code to GitHub, together with usage instructions and example data. See `https://doi.org/10.5281/zenodo.18095412`.

R1.35 **Comment:** L241: different results for some months is this random or is there some seasonality?

**Response**: We could not detect a seasonality in this case. Ockenfuß et al. [2025] and Kneifel and Moisseev [2020] investigated seasonality in more detail and over a longer timeseries and found generally a weak decrease in riming days from November to April. (see e.g. Fig. 6b in Ockenfuß et al. [2025]). However, the effect is too weak to explain the variability we see. Our results are likely dominated by natural variability.

**Technical Comments**

Thanks for pointing this out, all were corrected.

R1.36 **Comment:** L71: reference missing

R1.37 **Comment:** L91 (and later): remove the brackets around the year in the reference citep instead of (citet).

R1.38 **Comment:** L140: reference missing

R1.39 **Comment:** L201: Figure 5b is mentioned before Figure 5a please flip

**Response**: Corrected, also added a colorbar.

R1.40 **Comment:** Figure 4: please use color-blind friendly color maps.

R1.41 **Comment:** L236 (and likely earlier): Usually, Figure is written out at the beginning of a sentence and abbreviated to Fig. otherwise.

R1.42 **Comment:** L 270-275 (again way more lines than 5): the equations: I don't think all parameters are described.

R1.43 **Comment:** L369: reference missing

**Reviewer 2**

**General comments**

R2.1 **Comment:** The introduction would benefit from a stronger motivation for why detecting riming and understanding its drivers is important for cloud physics, precipitation processes, and numerical weather prediction.

**Response:** We restructured the introduction (see also the answer to item R1.1).

**Changes to the Manuscript:**

R2.2 **Comment:** use either the "German Meteorological Service" or "German Weather Service" consistently throughout the manuscript

**Response:** Corrected!

R2.3 **Comment:** as the other referee notes, your perspective is very Germany-centered and only in the last paragraph do you expand your view to the broader (European) perspective. Which other national meteorological services operate similar C-band networks? Do any routinely perform birdbath scans? Clarifying this would help position the work within a broader operational context.

**Response:** We restructured the introduction and also added this information in the conclusion.

R2.4 **Comment:** The original Kneifel & Moisseev (2020) retrieval used an average of MDV over a relatively long (several minutes) window to get rid of turbulence effects and wavy structures. How does this scale with the 15 rays every 5 minutes data from the C-Band radar network? Maybe I missed it in the text, but I think there should be a brief description of the Kneifel & Moisseev (2020) retrieval, how it was improved in Ockenfuß et al. (2025) and how it is finally applied to the C-Band data. You could list (maybe in the table) how many MDV values are averaged for each of the radar data sets, if you use a boxcar filter, and which MDV-FR relation from Kneifel & Moisseev (2020) is used for which of the data sets.

**Response:** Thanks for pointing that out. Regarding the details of the 15 s averaging, please also see our answer to your comment in item R2.16. The ability to remove wavy structures in the presence of only 5 minute profiles was one of our initial concerns as well and is why we introduced the resampled Ka-band dataset in the first place. In Sec. 2.6 "Convection Filtering" in the manuscript, we therefore test the applicability of the MDV averaging filter as used in Kneifel and Moisseev [2020] to our 5 min time resolution. The original retrieval is summarized in Sec. 2.1 in the manuscript. We added more details to this section and also to the table 1 in the manuscript. We also added a link forward to the convection filtering section in Sec. 2.1, to point out to the reader that more details about the boxcar averaging will follow.

**Specific Comments**

R2.5 **Comment:** line 19f: Clarify whether research cloud radars scan in the vertical or whether they typically perform vertically pointing observations without scanning. I tihnk that "vertically pointing observations" (and not scans) are the standard measurement strategy.

**Changes to the Manuscript:** Clarified in the manuscript.

R2.6 **Comment:** what should be Eq. (1), the number is missing here for some reason: I think that the subscript "c" for "crystal" is potentially misleading, as "crystals" typically denote pristine individual ice particles. Since the retrieval mainly targets riming of snowflakes, consider using notation referring to "unrimed snow," e.g. us (as in Moisseev et al., 2017), or something similar refering to both unrimed aggregates and crystals.

**Response:** We made sure the numbers of equations are rendered correctly. We changed the subscript.

R2.7 **Comment:** l. 62: The first figure referenced is Fig. 7a,b, which disrupts the flow. Consider omitting the figure reference here and describing the retrieval conceptually, returning to the examples later.

**Response:** We moved the example description and figure reference to the results section.

R2.8 **Comment:** Figs. 3 and 4 and 7: use the same colormap for the Doppler velocity. I generally like the use of colorblind friendly color schemes as in Fig. 3. In this context, please also ensure Fig. 1 is accessible to readers with color-vision deficiencies.

**Response:** We adapted Fig. 1 to use not only different colors, but also different shapes for better colorblind friendliness. Please note that Fig. 3 is different from Fig. 4 and 7 in the sense that it focuses on the contrast rain-snow, instead of variability within the ice phase like Fig. 4 and 7. Fig. 3 also has different colorbar limits. Therefore, we decided to choose a different colorbar here. In Fig. 4 and 7 (and all other figures with rainbow colorschemes), we use the "turbo" colormap. "turbo" is developed by Google with better perceptual uniformity compared to the popular 'jet' colormap, which includes sudden brightness changes. "turbo" is also suited for almost all color-vision deficiencies. Please see this text for more information: `URLhttps://research.google/blog/turbo-an-improved-rainbow-colormap-for-visualization/`

R2.9 **Comment:** l. 62: Fig. 7a and b : Please provide site and instrument information for the frontal systems shown in the referenced figures in the text to improve understandability.

**Response:** Can you clarify which specific information you are missing? Maybe, this point is already solved by moving the reference and figure description in item R2.7 to the results section, where all methods and sites are already described.

R2.10 **Comment:**l. 71: there is a reference missing.

**Response:** Corrected!

R2.11 **Comment:** l. 74f: "In the vertical, the intrinsic data resolution is 60m, but sampling is performed at 25m resolution (Gergely et al., 2022)." Please clarify how a dataset with intrinsic 60 m resolution is sampled at 25 m, are spectra interpolated or is the FFT performed on overlapping parts of the time series? And does it make sense to look at the data in 25 m resolution then?

**Response:** Gergely et al. [2022] found that the 25 m resolution offers more spatial structure in the birdbath scan than the nominal 60 m resolution, even though the individual pulses are not completely independent samples. The oversampling is happening directly in the signal processor, implemented by the manufacturer (GAMIC GmbH). It is operationally performed for the German radar network.

R2.12 **Comment:**l. 92 : I doubt a bit that it is possible to apply the EDR retrieval by Borque et al. (2016) to the 5min resolution birdbath scan, even if you took relatively long (e.g. 20 minute) windows to apply the retrieval. I think it would still be too few data points to fit turbulence spectrum slopes reliably. Please check this, or remove the reference here.

**Response:** This is exactly the point we want to make here: Some analysis techniques require a certain minimum time resolution to work. Therefore, they might not be transferable to the C-band. You may be correct that Borque et al. [2016] are among those techniques. One easy way to find out would be to do exactly what we did: Coarse-bin cloud radar data (where you already know that EDR can be retrieved reliably), apply the retrieval to the resampled data and compare the results. This is why we included the citation here.

R2.13 **Comment:**l. 93 "equipped"

**Response:** Corrected!

R2.14 **Comment:**l. 102ff: Why are you addressing the points 1-3 in reverse order, i.e. start with the last point and then moving on to the second and finally the first? Consider changing the order and separating the considerations regarding each of the three points with a line break to improve readability.

**Response:** Changed accordingly!

R2.15 **Comment:**105ff: how are you upsampling the 35 m resolution MDV data to 25 m? Please state which kind of interpolation you are using in the text.

**Response:** We use linear interpolation and added this in the text.

R2.16 **Comment:**You are not taking every 10 th cloud radar data profile but every 10 th cloudnet profile, correct? I understand that cloudnet categorize data are already 30 s averages of the raw MDV: `https://github.com/actris-cloudnet/cloudnetpy/blob/main/cloudnetpy/categorize/radar.py#L347` I guess this has implications for the mockup data and all the error considerations downstream (possibly the kappa filter): If the mockup data are 30 second averages, you miss small-scale variability that you would have in unaveraged MIRA-35 data. The goal is to use data similar as possible to the C-Band observations. I strongly recommend having another close look at the mockup data and making sure that it is really comparable to the C-Band birdbath observations. My guess is that the number of individual samples, coherent and uncoherent averages, going into one time-height observation in the C-Band radar data should be as close as possible to the number of samples in one time-height observation of the mockup data. Are the C-Band data averaged over 15 s more similar to the 30 s cloudnet resolution or to the "raw" MIRA-35 observations? Or does it make sense to aggregate a couple of raw MIRA measurement time steps to get to the resolution of the C-Band 15-s-data? My guess would be that 30 seconds of MIRA observations contain more than your 15*1024 pulses. It might make sense to actually dig into the spectra and (visually) make sure that the data look similar.

**Response:** This is a very valid point we have not yet considered in detail. Yes, we use 30s average cloudnet data, since this is used by Kneifel and Moisseev [2020], required for the reference retrieval and consistently available for many years. The original radar data resolution varies between different sites and years. To investigate this further, we have downloaded the 'radar' cloudnet product for one of our example cases from Fig. 7 in the manuscript, which contains the data in the original 10s resolution. In the left column of Figure 7 in this document, we average the original data (panel a) ) to 20s and 30s resolution. Then, we coarse sample the data in the right column to 5 min resolution by taking only every 30th, 20th and 10th sample from the left column, respectively. As we can see, the results are very similar. The dominating effect is the difference between the left and right column (i.e. the 5 min resolution), not the differences between the rows (pre-averaging the original data). The detected riming regions are also similar with respect to FR, time and height.

However, as you point out correctly, the difference between cloud radar and C-band radar is not only in the averaging time, but also in the number of averaged pulses, since the C-band radar operates at 1kHz pulse repetition frequency and the cloud radar operates at 5 kHz. So which aspect is more important: getting the number of pulses correct (15*1024) or the time interval (15s)? This question is hypothetical for our use case, since even the Cloudnet radar product in 10s resolution averages already 51200 pulses, so we can not go below that limit.

[Figure]

Figure 7: Effect of different averaging and coarse binning strategies. a),c),e): The original data in 10 s resolution and averaged to 20s and 30s, respectively. b),d),f): Coarse-binned version of a),c),e), by taking every 30th, 20th, and 10th profile to obtain a 5 min resolution.

Nevertheless, it is worth to investigate this question. Using the Mira35 radar in Munich, we performed a measurement during a recent snowfall event (Dec. 05, 2025). Fall speeds are around 1.5 m/s near the surface, indicating that there was at most some light riming present ($FR <= 0.6$). This is consistent with visual analysis of the snowflakes outside. We set the averaging time to 0.2s, equal to 1024 pulses. For this case, we can simulate the C-band radar by averaging every 5th sample over 15 s, therefore matching both, the 15s time interval and the number of pulses (15360). Figure 8b shows the resulting Doppler spectra. Figure 8c shows the same case, totally averaged over 15 s (75 samples and 76800 pulses). Figure 8d averages over 3 s (5 samples, 15360 pulses). As we can see, the spectra are very similar. To better quantify differences, we calculated the RMSE with respect to Figure 8b (using log-units). It turns out that, if one has to decide for one option, (approximately) matching the time interval is more important than matching the number of pulses. However, for practical purpose, the differences are small.

R2.17 **Comment:**For creating the mockup data, why didn't you take into account the lower sensitivity of the C-Band radar and the blind range, i.e. artificially worsen the Ka-Band data? I think it wouldn't be a big deal to exclude pixels below the height-dependent Ze threshold, or, for selected cases such as in Fig. 7, even go into the Doppler spectra and cut them at a higher

[Figure]

Figure 8: (a) Time-height view of radar reflectivity factor recorded with the collocated Metek X-band zenith radar (the Ka-band radar shows more gaps due to scans and changing radar settings) (b) Spectra, recorded with a Mira35 Ka-band cloud radar at 0.2 s averaging interval. The shown spectra are an average over 15 x 0.2 s samples, spaced equidistantly over a 15 s time interval. (c) Same as (b), but averaged over 75 samples within a 15 s time interval. (d) Same as (b), but averaged over 15 samples within a 3 s time interval.

threshold and then recompute MDV to see what happens when you miss all the liquid water peaks.

**Response:** Please see our combined answer to this question and the similar question of reviewer one, contained in item R1.2

R2.18 **Comment:**l. 110ff: I do not fully understand the meaning of these sentences. Please clarify: Are you saying that you are using the fit coefficients by Kneifel and Moisseev (2020) derived for X-Band on your C-Band data (but probably the ones for Ka-band on the mockup data?) and that X-Band and C-Band are sufficiently close together, the transition size where Mie effects play a role in X-Band would be 1 cm, and this can be neglected except for very large aggregates? You could add the information which FR-MDV relation is used, to Table 1.

**Response:** Yes. We added the line to Table 1 and additionally clarified this in the text.

R2.19 **Comment:**l. 131 "for most of the height range"

**Changes to the Manuscript:** Corrected.

R2.20 **Comment:**l. 140: missing reference.

**Response:** Corrected.

R2.21 **Comment:** The Cloudnet melting layer detection uses a combination of a peak in LDR (is this available for your cloudnet data?) and the criterion of MDV above 1 m/s at melting layer base, and, as a fallback if no LDR is available, only profiles of MDV, and, if available, spectrum width. This should be stated somewhere in the text.

**Response:** We added this information to the manuscript. Unfortunately, LDR is not available for the operational C-band birdbath scan. We also added this fact to the manuscript.

R2.22 **Comment:**Why develop a new algorithm instead of using or extending the Cloudnet MDV/ width one? In general though, it looks like the melting layer detection method proposed here seems to work very well, and this might also be of interest for Cloudnet developers. This could motivate a small comparison study, like in Fig. 3. The cloudnet melting layer code detection is here: `https://github.com/actris-cloudnet/cloudnetpy/blob/main/cloudnetpy/categorize/melting.py#L128`

**Response:** Thanks, a more detailed comparison would be indeed interesting! Initially, we were uncertain about which data sources (MDV, LDR, other instruments?) go into the Cloudnet melting layer detection and how their availability influences the quality of the melting layer detection. Therefore, we decided to develop our own approach, tailored to the C-band radars. It would be very interesting to combine maybe our idea of "maximum difference in averages above/below ML" with the Cloudnet implementation.

R2.23 **Comment:**l. 157: fallspeeds (lower case) and "criterion" instead of "criteria"

**Response:** Corrected!

R2.24 **Comment:**l. 174: "In Fig. 2c, only winter months (Nov to Apr)... ": this sentence is somewhat out of context here. I would mention this earlier, when discussing Fig. 2 (without having explained the melting layer detection yet but referring the reader to the next section), and maybe also write it in the caption of Fig. 2.

**Response:** Apologies, this was a false reference. This should be Fig. 3c.

R2.25 **Comment:**Side question: Why consider only winter months? You are mentioning it later in the Discussion, that it is to exclude convection, but I think this should be explained much earlier.

**Response:** We added an explanation in the methods section when discussing the issue of convection filtering. See also item R1.22.

R2.26 **Comment:** l. 176f: "Only if no reliable detection was possible, we extrapolate the last value up to 1 h and otherwise use the value from the nearest sounding station as a proxy. In the latter case, at most 12 h are tolerated between sounding and radar profile." How far are the nearest sounding stations for the different radars?

**Response:** In the map in Fig. 1, we can see the location of the sounding stations and radars. The biggest distance is between Feldberg (fbg) and Stuttgart (138 km) and between Dresden (drs) and Lindenberg (123 km). On average, the distance is 70 km.

R2.27 **Comment:** l. 186: replace the "original cloud radar resolution" with the cloudnet categorize/ classification time resolution or something similar

**Response:** Adapted!

R2.28 **Comment:** approx. l. 200 (line numbering is broken there): collocated, not collocated

**Response:** Corrected!

R2.29 **Comment:** Fig. 5a and b are referenced in opposite order in the text, consider switching them

**Response:** Corrected!

R2.30 **Comment:** l. 204: 20,000 instead of 20.000

**Response:** Corrected!

R2.31 **Comment:** l. 212 remove duplicate "percent"

**Response:** Corrected!

R2.32 **Comment:** l. 222ff: Your point 3 is scientifically meaningful and worth further discussion.

**Response:** Yes, we will discuss them in Sec. 3.3 and show that they are linked to precipitation patterns.

R2.33 **Comment:** Fig. 6: Maybe I missed it somewhere in the text, but I am missing a definition of "winter" and a discussion why these months were chosen and not differently. Sometimes "winter" refers to the months of DJF only, but it is different in your case. In one place, you write November to April, in another "14 months between October 2021 and April 2023".

**Response:** Yes, we define it as November to April. The reason is that heavily convective thunderstorms are much less likely during this time. See, for example, Fig. 11 in Taszarek et al. [2019]. We decided to restrict to those months to be maximally conservative in the application of our retrieval and evaluation of the results, even though we implemented filters to detect and remove heavily convective time steps.

**Changes to the Manuscript:** "14 months .." was corrected to "12 months between November 2021 and April 2023". We also added an explanation to the methods section in the convection filtering part.

R2.34 **Comment:** Also Fig. 6: Even though it's kind of obvious, I would also mention why there is a vertical dashed line at 0 degrees (no riming at higher temperatures)

**Response:** We added a legend element for the vertical dashed line.

R2.35 **Comment:** Fig. 7: This is a matter of taste, but I think it would benefit from adding the site and date as column headers.

**Response:** We added column headers for the date. The site is indicated by the row headers.

R2.36 **Comment:** beginning of Section 3.3 (the line numbers are broken again): I would mention the second outlier, Neuhaus, much earlier in the text. You mention that the Feldberg site is influenced by orography. Any ideas why the Neuhaus site sticks out? Interestingly, it does not stick out with respect to temperature distribution of the riming onset in Fig. 13. Any ideas why?

**Response:** Neuhaus could also be influenced by orography. The radar is located in the Thuringian Forest, another German mountain range of medium elevation (radar altitude: 880 m amsl). However, the variability of the surrounding terrain is not as strong as for the Feldberg site and the radar is located further inward the mountain range, farther away from the initial rise. This fits very well with the "intermediate" position in the riming and precipitation statistics between most other radars and the Feldberg. Likely, the more moderate terrain compared to the Feldberg radar is also the cause for the missing temperature shift. Clearly, the causing factor behind the shift is missing. As discussed, this factor could either be the presence of persistent vertical air motion or a difference in riming microphysics, which we can not easily distinguish from radar Doppler observations alone.

**Changes to the Manuscript:** We added the information above about the Neuhaus site to the discussion.

R2.37 **Comment:** last 2 lines before the equation: For better readability, I would refer to the letters in the equation in the explanatory text, i.e. "we decompose the total precipitation per site, T, into the product of the number of hours with precipitation, N and the average hourly precipitation rate R:

**Response:** Good recommendation, adapted!

R2.38 **Comment:** l. 272ff: Equation numbers are missing again.

**Response:** Corrected!

R2.39 **Comment:** Fig. 10: In 10b, it's not so easy to see the wet south and dry northeast. Consider changing the colorbar a bit

**Response:** Maybe the adapted legend/colorbar with the circle sizes also included helps?

R2.40 **Comment:** l 273ff: The discussion of Figs. 11 and 12 could use a bit more detail. I think these are some of the main results in this paper and I find it a bit hard to follow. In particular Fig. 12b is only discussed very briefly.

**Response:** Are there any specific details you would like to be discussed?

R2.41 **Comment:** l 290: "who" instead of "which"

**Response:** Corrected!

R2.42 **Comment:** l. 316ff: vertical air motions: The kappa filter removes updrafts, but may leave stationary or slowly varying downdrafts unflagged. Stationary downdrafts could be misinterpreted as riming by the algorithm. I guess that those downdrafts however would not shift the entire temperature distribution in Fig. 13 to lower temperatures for the Feldberg site, as you observe, but might increase riming frequency equally across most of the temperature range (if we assume that the downdrafts influence the entire atmospheric column similarly). In connection with the lower number of observations at higher temperatures due to higher elevation of the Feldberg site, it's a bit hard to wrap the head around the different effects different factors could have on the temperature distribuion curve. While for the other (not orography influenced sites) I would expect a probability of 0 at very low temperatures, say below -20°C, at the Feldberg site, there could be also riming at temperatures below that range, maybe even below -40°C, due to misclassifications, e.g. if the entire atmospheric column is in a downdraft for several minutes. It is somewhat eye-catching that fbg is the only curve extending down to the x-axis lower limit, and maybe even beyond. How does the curve look like for lower temperatures below the x-axis lower limit in Fig. 13? This can give a hint on the uncertainty of the method introduced by orography.

**Response:** Yes, your thoughts are very similar to our thoughts at this point of the research. As you note correctly, one would expect a more uniform temperature distribution, if we would observe only vertical air motions instead of riming. When interpreting the temperature distribution in Fig. 13, you should be able to ignore the fact that the number of observations is lower for the higher temperatures at Feldberg, since we corrected the curves for this effect (see also Sec. 2.9 "Histogram Correction"). In Fig. C1 of the appendix, we added the uncorrected and unnormalized curves for reference. We also extended the x-axis lower limit of Fig. 13.

R2.43 **Comment:** If you are not normalizing to an area of 1 below the curve but rather look at the likelihood of riming per temperature bin as Kneifel & Moisseev (2020) did in their Fig. 8, how does fbg then compare to the other sites?

**Response:** Also motivated by item R1.27, we introduced a new appendix section, including the figure with absolute counts and a statistical analysis of the curves.

R2.44 **Comment:** l. 331ff: the idea that stronger updrafts might be responsible for the shift in the temperature distribution towards colder temperatures at first sounds plausible, but aren't these the cases that the kappa filter flags as convective/ updraft?

**Response:** Not necessarily. An updraft of e.g 50 cm/s with rimed particles of 2 m/s fallspeed will still produce a Doppler signal of 1.5 m/s downward. In this sense, updrafts will cause us to detect less or weaker riming, so if there is a bias in riming detection, it will more likely be underestimation than overestimation. Of course, if the updraft is that strong that we see actual upward movement of the particles, this is rightfully excluded by the filter. Please also note that the airmass lifting might not necessarily be located at the same position as the falldown of rimed particles. The Feldberg radar is still located about 20 km east of the western edge of of the blackforest mountain range, where the terrain starts to rise.

R2.45 **Comment:** did you look into whether there is a correlation of FR and precipitation intensity?

**Response:** No, for this part of the analysis, we only considered the presence or absence of strong riming (i.e. above our detection limit of FR=0.6). Please see Ockenfuß et al. [2025], who looked more into the distribution and vertical profiles of FR.

R2.46 **Comment:** Conclusions: One point from earlier in the manuscript could be brought up again here: Saving high-resolution model profiles of T, p, q and wind for all the weather radar sites should not be a big deal as they are also being saved for Cloudnet sites, correct?

**Response:** Yes! At least, we think so. We added this to the conclusion.

R2.47 **Comment:** In general, the Conclusions are beautifully written, and motivating.

**Response:** Thank you, this is really nice to hear!

**References**

P. Borque, E. Luke, and P. Kollias. On the unified estimation of turbulence eddy dissipation rate using doppler cloud radars and lidars. *Journal of Geophysical Research: Atmospheres*, 121(10): 5972–5989, May 2016. ISSN 2169-8996. doi: 10.1002/2015jd024543.

M. Gergely, M. Schaper, M. Toussaint, and M. Frech. Doppler spectra from dwd's operational c-band radar birdbath scan: sampling strategy, spectral postprocessing, and multimodal analysis for the retrieval of precipitation processes. *Atmospheric Measurement Techniques*, 15(24):7315–7335, Dec. 2022. ISSN 1867-8548. doi: 10.5194/amt-15-7315-2022.

R. Hogan and E. Connor. Facilitating cloud radar and lidar algorithms: the cloudnet instrument synergy/target categorization product. Sept. 2004. URL https://www.met.reading.ac.uk/~swrhgnrj/publications/categorization.pdf.

S. Kneifel and D. Moisseev. Long-term statistics of riming in nonconvective clouds derived from ground-based doppler cloud radar observations. *Journal of the Atmospheric Sciences*, 77(10): 3495–3508, 2020. doi: 10.1175/JAS-D-20-0007.1. URL https://journals.ametsoc.org/view/journals/atsc/77/10/jasD200007.xml.

P. Ockenfuß, M. Gergely, M. Frech, and S. Kneifel. Spatial and temporal scales of riming events in nonconvective clouds derived from long-term cloud radar observations in germany. *Journal of Geophysical Research: Atmospheres*, 130(4), Feb. 2025. ISSN 2169-8996. doi: 10.1029/2024jd042180.

M. Taszarek, J. Allen, T. Púčik, P. Groenemeijer, B. Czernecki, L. Kolendowicz, K. Lagouvardos, V. Kotroni, and W. Schulz. A climatology of thunderstorms across europe from a synthesis of multiple data sources. *Journal of Climate*, 32(6):1813–1837, Mar. 2019. ISSN 1520-0442. doi: 10.1175/jcli-d-18-0372.1.

D. Wolfensberger, D. Scipion, and A. Berne. Detection and characterization of the melting layer based on polarimetric radar scans. *Quarterly Journal of the Royal Meteorological Society*, 142 (S1):108–124, Nov. 2015. ISSN 1477-870X. doi: 10.1002/qj.2672.